# An optimized quantitative proteomics method establishes the cell type-resolved mouse brain secretome

Johanna Tüshaus[1,2] ID, Stephan A Müller[1,2], Evans Sioma Kataka[3], Jan Zaucha[3], Laura Sebastian Monasor[1], Minhui Su[1,4] ID, Gökhan Güner[1,2], Georg Jocher[1,2], Sabina Tahirovic[1] ID, Dmitrij Frishman[3], Mikael Simons[1,4,5] & Stefan F Lichtenthaler[1,2,5,*] ID

## Abstract

To understand how cells communicate in the nervous system, it is essential to define their secretome, which is challenging for primary cells because of large cell numbers being required. Here, we miniaturized secretome analysis by developing the "high-performance secretome protein enrichment with click sugars" (hiSPECS) method. To demonstrate its broad utility, hiSPECS was used to identify the secretory response of brain slices upon LPS-induced neuroinflammation and to establish the cell type-resolved mouse brain secretome resource using primary astrocytes, microglia, neurons, and oligodendrocytes. This resource allowed mapping the cellular origin of CSF proteins and revealed that an unexpectedly high number of secreted proteins *in vitro* and *in vivo* are proteolytically cleaved membrane protein ectodomains. Two examples are neuronally secreted ADAM22 and CD200, which we identified as substrates of the Alzheimer-linked protease BACE1. hiSPECS and the brain secretome resource can be widely exploited to systematically study protein secretion and brain function and to identify cell type-specific biomarkers for CNS diseases.

**Keywords** BACE1; brain cells; CSF; proteomics; secretomics
**Subject Categories** Neuroscience; Proteomics
**The EMBO Journal (2020) 39: e105693**

## Introduction

Protein secretion is essential for inter-cellular communication and tissue homeostasis of multicellular organisms and has a central role in development, function, maintenance, and inflammation of the nervous system. Proteins secreted from cells are referred to as the secretome and comprise secreted soluble proteins, such as insulin, granulins, APOE, and extracellular matrix proteins (e.g., neurocan, fibronectin). The secretome also comprises the extracellular domains of membrane proteins, e.g., growth factors, cytokines, receptors, and cell adhesion proteins (e.g., neuregulin, NCAM, N-cadherin), which are proteolytically generated by mostly membrane-bound proteases and secreted in a cellular process called ectodomain shedding (Lichtenthaler et al, 2018). However, it is largely unknown to what extent ectodomain shedding contributes to total protein secretion and how this differs between cell types in the brain.

Omics' approaches have generated large collections of mRNA and protein abundance data across the different cell types of the brain (e.g., Zhang et al, 2014; Sharma et al, 2015). In contrast, little is known about the proteins that are secreted from brain cells and whether—in parallel to their broad expression in different brain cell types—they are secreted from multiple brain cell types or instead are secreted in a cell type-specific manner *in vitro, ex vivo* (e.g., organotypic slice culture), *and in vivo*. Cerebrospinal fluid (CSF) constitutes an *in vivo* brain secretome and is an easily accessible body fluid widely used for studying brain (patho-)physiology and measuring and identifying disease biomarkers (Olsson et al, 2016; Johnson et al, 2020; Zetterberg & Bendlin, 2020), but it is largely unknown which cell type the CSF proteins are secreted from, because no systematic brain cell type-specific protein secretion studies are available.

Dysregulated protein secretion and shedding are linked to neurologic and psychiatric diseases, including neurodegeneration, e.g., APP, APOE, SORL1, and TREM2 in Alzheimer's disease (AD), or the prion protein (PRNP) in prion disease (Lichtenthaler et al, 2018). Thus, the identification and quantification of secretomes do not only allow understanding of biological processes under physiological conditions, but also contribute to unraveling the molecular basis of diseases and identification of drug targets and biomarkers, such as shed TREM2 for AD (Suarez-Calvet et al, 2016; Ewers et al, 2019; Schindler et al, 2019).

1   German Center for Neurodegenerative Diseases (DZNE), Munich, Germany
2   Neuroproteomics, School of Medicine, Klinikum rechts der Isar, Technical University of Munich, Munich, Germany
3   Department of Bioinformatics, Wissenschaftszentrum Weihenstephan, Technical University of Munich, Freising, Germany
4   Institute of Neuronal Cell Biology, Technical University Munich, Munich, Germany
5   Munich Cluster for Systems Neurology (SyNergy), Munich, Germany
    *Corresponding author. Tel: +49 89 440046425; E-mail: stefan.lichtenthaler@dzne.de

Systematic identification and quantification of secretome proteins are commonly done using conditioned medium of a (primary) cell type and its analysis by mass spectrometry-based proteomics. A major challenge is the low concentration of secreted proteins within the conditioned medium (Schira-Heinen *et al*, 2019). Therefore, many studies concentrate medium from tens of millions of cells (Kleifeld *et al*, 2010, 2011; Kuhn *et al*, 2012; Wiita *et al*, 2014; Schlage *et al*, 2015). However, such numbers are often not available for primary cells, such as microglia, where on average one million cells may be purified from the brain of individual adult mice. Thus, miniaturized secretome analysis methods are required. A second major challenge is the large dynamic range of secretomes, in particular when cells are cultured in the presence of serum or serum-like supplements, which are highly abundant in proteins (most notably albumin), hampering the detection of endogenous, cell-derived secreted proteins, whose protein levels are typically orders of magnitude lower (Eichelbaum *et al*, 2012). Therefore, cells are often cultured under serum- and protein-free starvation conditions (Kleifeld *et al*, 2010; Meissner *et al*, 2013; Deshmukh *et al*, 2015), which, however, can strongly alter secretome composition and may induce cell death (Eichelbaum *et al*, 2012). An alternative approach, compatible with cell culture in the presence of serum or serum-like supplements, is to metabolically label the cell-derived but not the exogenous serum proteins with analogs of methionine or sugars that are incorporated into the protein backbone or glycan structures, of newly synthesized cellular proteins, respectively (Eichelbaum *et al*, 2012; Kuhn *et al*, 2012). However, even these established methods still require extensive secretome fractionation, either at the protein level or at the peptide level (Eichelbaum *et al*, 2012; Kuhn *et al*, 2012) and, thus result in laborious sample preparation, extensive mass spectrometry measurement times, and the requirement of large amounts of samples, which may not be available from primary cells or tissues.

Here, we developed the "high-performance secretome protein enrichment with click sugars" (hiSPECS) method, which downscales and speeds up secretome analysis and now allows secretome analysis of primary brain cells from single mice cultured in the presence of serum proteins. We applied hiSPECS to determine the cell type-resolved mouse brain secretome, which establishes a resource for systematically studying protein secretion and shedding in the brain. Broad applicability of hiSPECS and the resource is demonstrated (i) by gaining new insights into the extent of cell type-specific protein secretion and shedding; (ii) by identifying new substrates for the protease BACE1, a major drug target in AD; (iii) by determining the cellular origin of proteins in CSF; and (iv) by revealing that LPS-induced inflammatory conditions in organotypic brain slices do not only lead to inflammatory protein secretion from microglia, but instead induce to a systemic secretory response from multiple cell types in brain slices.

## Results

### Development of hiSPECS and benchmarking against SPECS

To enable secretome analysis of primary brain cell types from single mice under physiological conditions (i.e., in the presence of serum-like supplements), we miniaturized the previously established SPECS method, which required 40 million cells per experiment (Kuhn *et al*, 2012). We introduced four major changes (Fig 1A; see Fig EV1A for a detailed comparison of hiSPECS versus SPECS). First, after labeling of cells with N-azido-mannosamine (ManNAz), an azido group-bearing sugar, secretome glycoproteins were enriched from the conditioned medium with lectin-based precipitation using concanavalin A (ConA). This strongly reduced albumin, which is not a glycoprotein (Fig EV1B). Because the majority of soluble secreted proteins and most of the membrane proteins—which contribute to the secretome through shedding—are glycosylated (Kuhn *et al*, 2016), hiSPECS can identify the major fraction of secreted proteins. Second, we selectively captured the azido-glycoproteins by covalent binding to magnetic dibenzylcyclooctyne (DBCO)–alkyne beads using copper-free click chemistry. This allowed stringent washing to reduce contaminating proteins. Third, on-bead digestion of the captured glycoproteins was performed to release tryptic peptides for mass spectrometry-based label-free protein quantification (LFQ). Fourth, mass spectrometry measurements were done on a Q Exactive HF mass spectrometer using either data-dependent acquisition (DDA) or the more recently developed data-independent acquisition (DIA) (Bruderer *et al*, 2015; Gillet *et al*, 2012; Ludwig *et al*, 2018; Fig EV1C). DDA and DIA are two different modes in which proteomic data can be acquired. In contrast to DDA, DIA is not limited to the most abundant peptides for the subsequent peptide sequencing and identification, but fragments, and detects all peptides within a defined m/z window, which may be particularly advantageous for the detection of lower abundant peptides (Gillet *et al*, 2012; Bruderer *et al*, 2015; Ludwig *et al*, 2018; Sebastian *et al*, 2020). To benchmark hiSPECS against the previous SPECS protocol, we collected the secretome of primary murine neurons in the presence of a serum supplement as before (Kuhn *et al*, 2012), but used only one million neurons (40-fold fewer cells compared with SPECS). Despite this miniaturization, hiSPECS quantified on average 186% and 236% glycoproteins in DDA and DIA modes, respectively, compared with the previous SPECS dataset (Kuhn *et al*, 2012) (according to UniProt and four previous glycoproteomic studies; Figs 1B and EV1D and E; Table EV1) (Zielinska *et al*, 2010; Fang *et al*, 2016; Liu *et al*, 2017; Joshi *et al*, 2018). Furthermore, the DIA method provided 18% more quantified glycoproteins and 11% shed transmembrane proteins compared with DDA (Fig 1C). Overall, DIA extended the dynamic range for secretome protein quantification by almost one order of magnitude, which was evaluated based on intensity-based absolute quantification (iBAQ) values representing a rough estimate of molar protein abundances within a sample (Schwanhäusser *et al*, 2011; Fig 1D). Due to the superiority of DIA over DDA in regard of proteome coverage and reproducibility for secretome analysis, we focused on hiSPECS DIA throughout the manuscript. 99.9% (2,273/2,276) of unique tryptic peptides identified from single-pass transmembrane proteins mapped to their known or predicted protein ectodomains. This is an important quality control, which demonstrates that the secretome contains proteolytically shed transmembrane protein ectodomains and not full-length transmembrane proteins and that hiSPECS reliably identifies secretome-specific proteins (Fig 1E).

The novel hiSPECS procedure does not require further protein or peptide fractionation. Thus, sample preparation time and mass spectrometer measurement time were both reduced fivefold compared with previously required times (Fig 1F). Importantly, hiSPECS also improved the reproducibility of protein LFQ among different biological replicates, which is reflected by an average Pearson correlation

coefficient of 0.97 using hiSPECS in comparison with 0.84 with the previous procedure (Fig EV1F). Taken together, hiSPECS outperforms SPECS with regard to the required number of cells, sensitivity, reproducibility, secretome coverage, sample preparation, and mass spectrometry measurement time.

## Cell type-resolved mouse brain secretome resource

Secreted soluble and shed proteins have key roles in signal transduction, but their cellular origin is often unclear as they may be expressed by multiple cell types. Thus, we used hiSPECS to

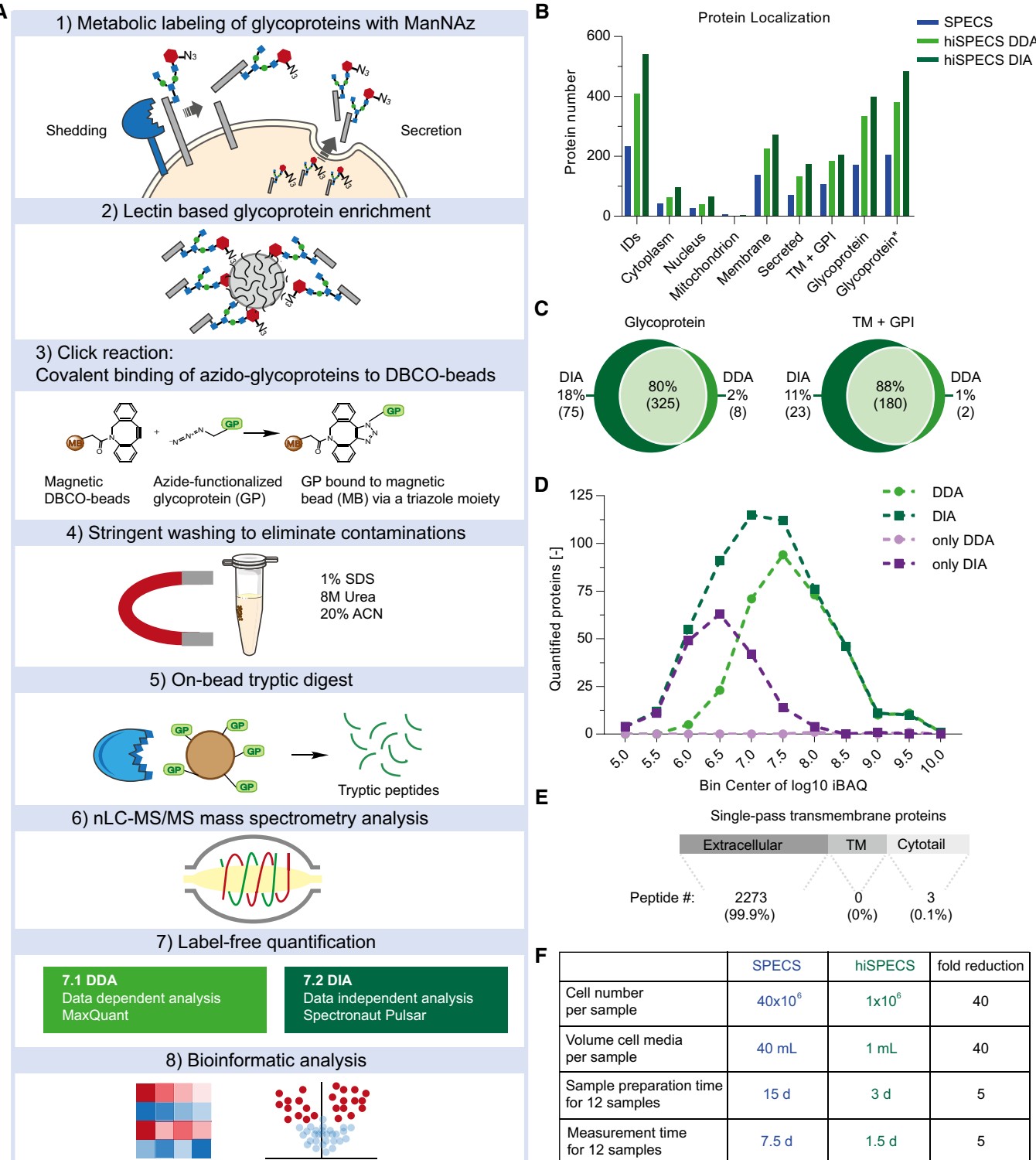

**Figure 1.**

**Figure 1.   Workflow of the hiSPECS method and benchmarking against SPECS.**

A   Graphical illustration of the hiSPECS workflow. Cells are metabolically labeled with N-azido-mannosamine (ManNAz), an azido group-bearing sugar, which is metabolized in cells and incorporated as azido-sialic acid into N- and O-linked glycans of newly synthesized glycoproteins, but not into exogenously added serum proteins. ACN: acetonitrile.

B   Bar chart indicating protein quantification, protein localization (according to UniProt) in the secretome of primary neurons, comparing the previous SPECS (blue) to the new hiSPECS method using DDA (light green) or DIA (dark green). Proteins were counted if quantified in at least 9 of the 11 biological replicates of hiSPECS or 4 of 5 biological replicates of the previous SPECS paper (Kuhn *et al*, 2012). The category glycoprotein* includes UniProt annotations and proteins annotated as glycoproteins in previous papers (Zielinska *et al*, 2010; Fang *et al*, 2016; Liu *et al*, 2017; Joshi *et al*, 2018). Proteins can have multiple UniProt annotations, e.g., for APP membrane, TM, cytoplasm, and nucleus, because distinct proteolytic fragments are found in different organelles, so that some proteins in categories cytoplasm and nucleus may overlap with the categories secreted and TM + GPI. IDs: identified proteins.

C   Venn diagram comparing the number of protein groups quantified (5/6 biological replicates) by DDA versus DIA of the same samples. Left panel: glycoprotein. Right panel: TM and GPI proteins, which are potentially shed proteins.

D   Distribution of quantified proteins with DDA and DIA (at least in 3 of 6 biological replicates). The number of quantified proteins is plotted against the $\log_{10}$-transformed intensity-based absolute quantification (iBAQ) values with a bin size of 0.5. The iBAQ values roughly correlate with molar abundance of the proteins. Therefore, a difference in one in $\log_{10}$ scale represents a 10-fold abundance difference. The number of quantified proteins per bin for DDA and DIA is indicated in light and dark green, respectively. Proteins that were only quantified with DDA or DIA are colored in light and dark purple, respectively.

E   All tryptic peptides identified from TM proteins were mapped onto their protein domains. Only 0.1% of the peptides mapped to intracellular domains, demonstrating that secretome proteins annotated as TM proteins comprise the shed ectodomains, but not the full-length forms of the proteins.

F   Comparison of SPECS and hiSPECS method with regard to cell number, volume of culture media, sample preparation time, and mass spectrometer measurement time.

Data information: TM: single-pass transmembrane protein; GPI: glycosylphosphatidylinositol-anchored membrane protein.

establish a resource of the brain secretome in a cell type-resolved manner, focusing on the four major brain cell types—astrocytes, microglia, neurons, and oligodendrocytes (Fig 2A; Table EV2, Appendix Fig S1). One million primary cells of each cell type were prepared from individual mouse brains. For further analysis, we focused on proteins detected in the secretome of at least five out of six biological replicates of at least one cell type. This yielded 995 protein groups in the secretome, with microglia having on average the largest (753) and astrocytes the smallest (503) number of protein groups (Fig 2B). GO cellular compartment analysis revealed extracellular region to be the most enriched term underlining the quality of our secretome library (Fig EV2A and B). An additional quality control measure was the enrichment of known cell type-specifically secreted marker proteins such as LINGO1, SEZ6, and L1CAM in the neuronal secretome (Fig EV2C). Importantly, the secretome analysis identified 111 proteins (Fig 2C) that were not detected in a previous proteomic study (Sharma *et al*, 2015; Table EV3), that identified > 10,000 proteins in the lysates of the same primary mouse brain cell types, which had been taken in culture as in our study (Table EV2). This includes soluble proteins (e.g., CPN1, TIMP1) and shed ectodomains (e.g., ADAM19, CRIM1, FRAS1) and even the protein GALNT18, which was assumed to be a pseudogene that is not expressed as a protein. Together, this demonstrates that secretome analysis is complementary to lysate proteomics in order to identify the whole proteome of an organ.

Based on $\log_2$ transformed LFQ intensity values, the Pearson correlation coefficients between the six biological replicates of each cell type showed on average an excellent reproducibility with a value of 0.92 (Fig 2D). In strong contrast, the correlation between different cell types was dramatically lower (0.29–0.68), indicating prominent differences between the cell type-specific secretomes (Figs 2D–F, EV2D, and EV3). In fact, about one-third (322/995) of the secretome proteins were consistently detected (5/6 biological replicates) in the secretome of only one cell type and in fewer replicates or not at all within the other cell type secretomes (Fig 2G), highlighting the unique cell type-characteristic secretome fingerprint of each cell type (Table EV4). Pairwise correlation revealed that

some secretomes correlate more closely than others (Fig 2D). For example, the secretome of oligodendrocytes correlated to a higher degree with the secretome of astrocytes than with the secretome of neurons, which was also observed for the cell lysate proteome of the same brain cell types (Sharma *et al*, 2015) and may reflect their common origin form the glia lineage (Hirano & Goldman, 1988).

Cell type-specific protein secretion was visualized in a heat map (Figs 3A and EV3A). Gene ontology analysis of the enriched secretome proteins revealed functional clusters corresponding to the known functions of the four individual cell types. For example, proteins related to the metabolic processes, gliogenesis, and immune response were preferentially secreted from astrocytes (e.g., IGHM, CD14, LBP). Proteins of the functional clusters autophagy and phagocytosis were secreted from microglia (e.g., TGFB1, MSTN, TREM2), in agreement with key microglial functions. Neuron-preferentially secreted proteins belonged to the neuron-specific categories axon guidance, trans-synaptic signaling, and neurogenesis (e.g., NCAN, CHL1, SEZ6). Oligodendrocyte-specifically secreted proteins (e.g., OMG, ATRN, TNFRSF21) fell into categories lipid metabolic process and myelination, in agreement with the role of oligodendrocytes in myelin sheet formation. This demonstrates that cell function can be obtained by a cell's secretome (Figs 3A and EV3B).

Secreted proteins may act as soluble cues to signal to other cells. To unravel the inter-cellular communication between secreted proteins and transmembrane proteins acting as potential binding partners/receptors, we mapped known interaction partners [from UniProt and BioGRID (Chatr-Aryamontri *et al*, 2015; The UniProt Consortium, 2018)], but now in a cell type-resolved manner (Fig 3B and Table EV5). Besides known cell type-specific interactions, such as between neuronally secreted CD200 and its microglia-expressed receptor CD200R1 (Yi *et al*, 2016), we also detected new cell type-specific interactions, e.g., between ADIPOQ (adiponectin) and CDH13. Adiponectin is a soluble anti-inflammatory adipokine with key functions not only in metabolism, but also in neurogenesis and neurodegeneration (Lee *et al*, 2019). Although adiponectin is thought to be secreted exclusively from adipocytes and assumed to reach the brain by crossing the blood–brain barrier, our resource

and the cell type-resolved interaction map reveal that adiponectin can also be produced and secreted from brain cells (oligodendrocytes). The binding to one of its receptors, cadherin 13 (neurons), establishes a new interaction between oligodendrocytes and neurons, which may have an important role in controlling the multiple, but not yet well-understood, adiponectin functions in the brain.

Besides pronounced cell type-specific protein secretion, another key insight obtained from our secretome resource is that ectodomain shedding of membrane proteins strongly contributes to the composition of the secretome (43%, 242/568 glycoproteins according to UniProt). The extent differed between the major mouse brain cell types (Fig 3C) and became even more evident when focusing on

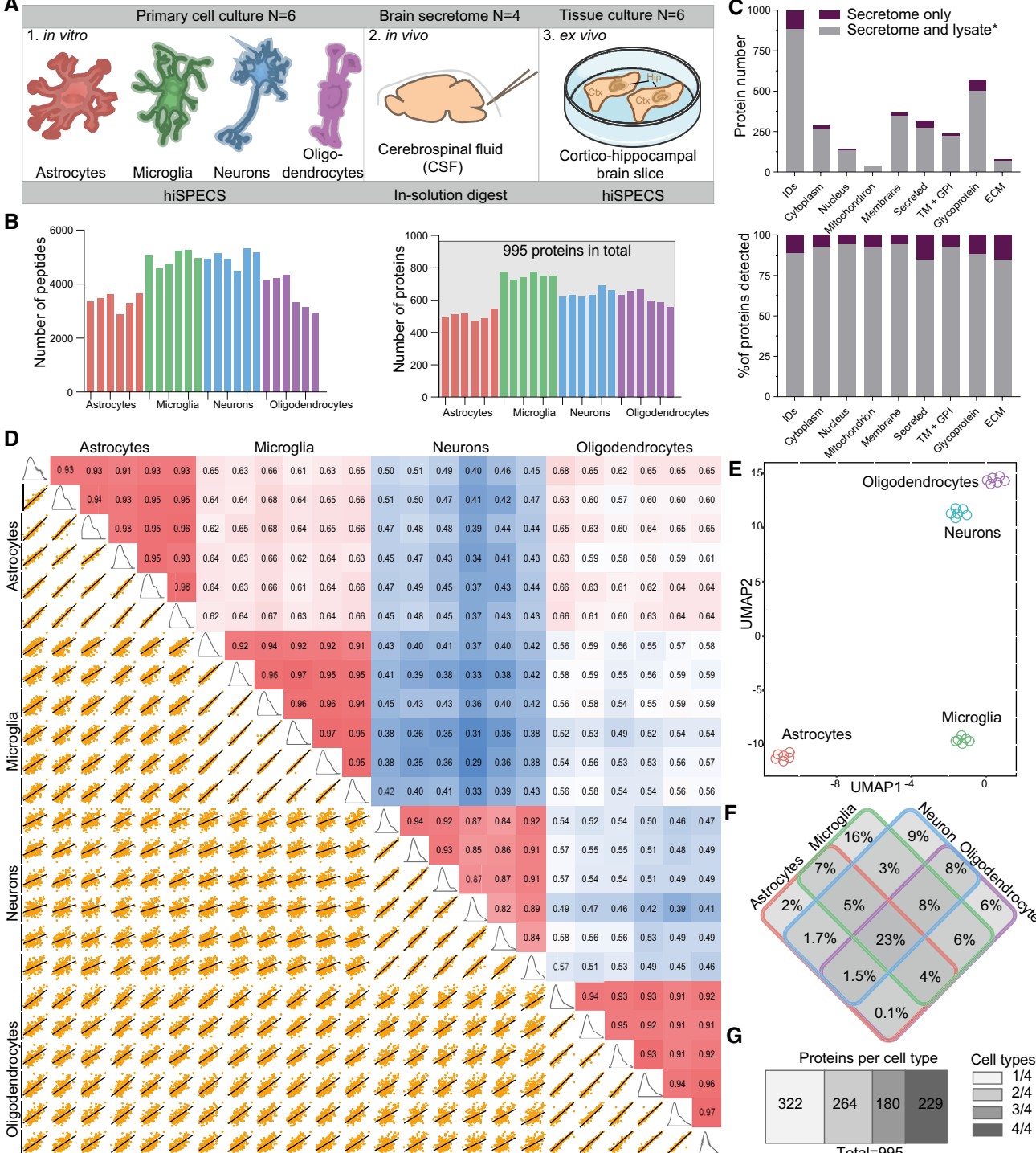

**Figure 2.**

**Figure 2. Cell type-resolved mouse brain secretome resource.**

A   Illustration of the proteomic resource including secretome analysis of primary murine astrocytes, microglia, neurons, oligodendrocytes, cerebrospinal fluid (CSF) analysis, and brain slices (hippocampus (Hip), cortex (Ctx)).

B   Number of peptides (left) and protein groups (right) quantified in the secretome of the investigated brain cell types with the hiSPECS DIA method. Proteins quantified in at least 5 out of 6 biological replicates in at least one cell type are considered. In total, 995 protein groups were detected.

C   995 proteins were quantified and identified (ID) in the hiSPECS secretome resource. The gray part of each column indicates how many proteins were also detected in the lysates of the same four cell types as analyzed in a previous proteome dataset (Sharma *et al*, 2015). This reveals that 111 proteins (purple) were only detected in the secretome. Relative to all the secretome proteins covered by the lysate study (gray), the most enriched UniProt annotation of the newly identified proteins (purple) was secreted and extracellular matrix (15%) (lower panel).

D   Correlation matrix showing the relationship between the different brain cell types. The matrix shows the Pearson correlation coefficient (red indicates a higher, blue a lower correlation) and the correlation plots of the $\log_2$ LFQ intensities of the secretome of astrocytes, neurons, microglia, and oligodendrocytes processed with the hiSPECS method.

E   UMAP (Uniform Manifold Approximation and Projection) plot showing the segregation of the brain cell type secretomes based on LFQ intensities of quantified proteins.

F   Venn diagram illustrating the percentage of the secretome proteins that were secreted from only one or multiple cell types. Proteins quantified in at least 5 biological replicates of one cell type were considered.

G   Bar graph indicating the number of protein groups, which were detected in one, two, three, or all cell types with at least 5 biological replicates.

the cell type-specifically secreted proteins. More than two-thirds (71%, 74/104) of the neuron-specifically secreted proteins were shed ectodomains (Fig 3D), i.e., proteins annotated as transmembrane or GPI-anchored proteins, of which we almost only (99.9%) identified peptides from the ectodomain (Fig 1E). The neuronally shed ectodomains contain numerous trans-synaptic signaling and cell adhesion proteins (NRXN1, SEZ6, CNTNAP4) indicating that shedding is an important mechanism for controlling signaling and synaptic connectivity in the nervous system. In contrast to neurons, shedding appeared quantitatively less important in astrocytes, where only 21% (9/43) of the cell type-specifically secreted proteins were shed ectodomains (Fig 3D). In contrast, soluble secreted proteins are particularly relevant for astrocytes and microglia where they constituted 70% (30/43) and 62% (38/61) of the secretome proteins, respectively, and contain numerous soluble proteins with functions in inflammation, e.g., complement proteins and TIMP1 (astrocytes) and GRN, MMP9, PLAU, and TGFB1 (microglia). Importantly, several soluble secreted proteins were expressed at similar levels in different brain cell types, but predominantly secreted from only one, suggesting that cell type-specific protein secretion may be an important mechanism to control brain inflammation. Examples are the astrocyte-secreted complement factor B and the microglia-secreted LTBP4 (Table EV4).

## Mechanisms of cell type-specific protein secretion

Abundance levels in the lysate of the majority of proteins are similar among astrocytes, microglia, neurons, and oligodendrocytes, with only around 15% of the proteins present in a cell type-specific manner (Sharma *et al*, 2015). In clear contrast, we observed that in the secretome of the same cell types, nearly half (420/995 proteins) of the secreted proteins were secreted in a cell type-specific manner (> 5-fold enriched in one secretome compared with all three others or consistently detected in the secretome of only one cell type; Table EV4), suggesting that cell type-specific protein secretion strongly contributes to functional differences between the four brain cell types.

An obvious mechanism explaining cell type-specific protein secretion is cell type-specific expression of the secreted soluble or shed proteins. Unexpectedly, however, a correlation with cell type-specific protein abundance in the same cell types (Sharma *et al*,

2015) was observed for only 20–30% of the cell type-specifically secreted proteins (Fig 4A), including the TGFβ coreceptor CD109 from astrocytes, the inflammatory proteins GRN, BIN2, and TGFβ1 from microglia, the cell adhesion proteins CD200, L1CAM, and the bioactive peptide secretogranin (CHGB) from neurons, and CSPG4, PDGFRα, OMG, and BCHE from oligodendrocytes (examples shown in Figs 4A and B, and EV4). This demonstrates that cell type-specific expression is only a minor or only one of several mechanisms controlling cell type-specific protein secretion and shedding. In fact, the vast majority of cell type-specifically secreted proteins were equally expressed in two or more cell types (Fig 4A).

Because we observed that shedding contributes significantly to protein secretion, particularly in neurons, we considered the possibility that cell type-specific protein shedding may mechanistically also depend on cell type-specific expression of the contributing shedding protease. For example, Alzheimer's disease-linked protease β-site APP-cleaving enzyme (BACE1) (Vassar *et al*, 1999; Yan *et al*, 1999), which has fundamental functions in the brain, is highly expressed in neurons, but not in astrocytes, microglia, or oligodendrocytes (Voytyuk *et al*, 2018). Consistent with our hypothesis, several known BACE1 substrates (APLP1, CACHD1, PCDH20, and SEZ6L), which are broadly expressed, were specifically shed from neurons (examples shown in Figs 4A and B, and EV4). To investigate whether additional proteins in our secretome resource may be shed by BACE1 in a cell type-specific manner, primary neurons (prepared from cortex plus hippocampus) were treated with the established BACE1 inhibitor C3 (also known as inhibitor IV; Stachel *et al*, 2004; Fig 5A; Table EV6), followed by hiSPECS. C3 can also inhibit BACE2, a close homolog of BACE1, but in contrast to BACE1, BACE2 is very little expressed in neurons (Voytyuk *et al*, 2018). This analysis revealed 29 transmembrane proteins with reduced ectodomain levels in the secretome (Figs 5B and EV5; Appendix Table S1), which were scored as BACE1 substrate candidates. Besides known substrates, hiSPECS identified additional BACE1 substrate candidates (ADAM22, CD200, CXADR, and IL6ST) in neurons (Fig 5B). Although BACE1 is broadly expressed in mouse brain, it is particularly highly expressed in hippocampus (Vassar *et al*, 1999). Thus, we first compared the secretome of cortical versus hippocampal neurons (Fig EV5C) to analyze whether the neuronal secretome differs between anatomical brain regions. Second, we repeated the BACE1 inhibition experiment with pure

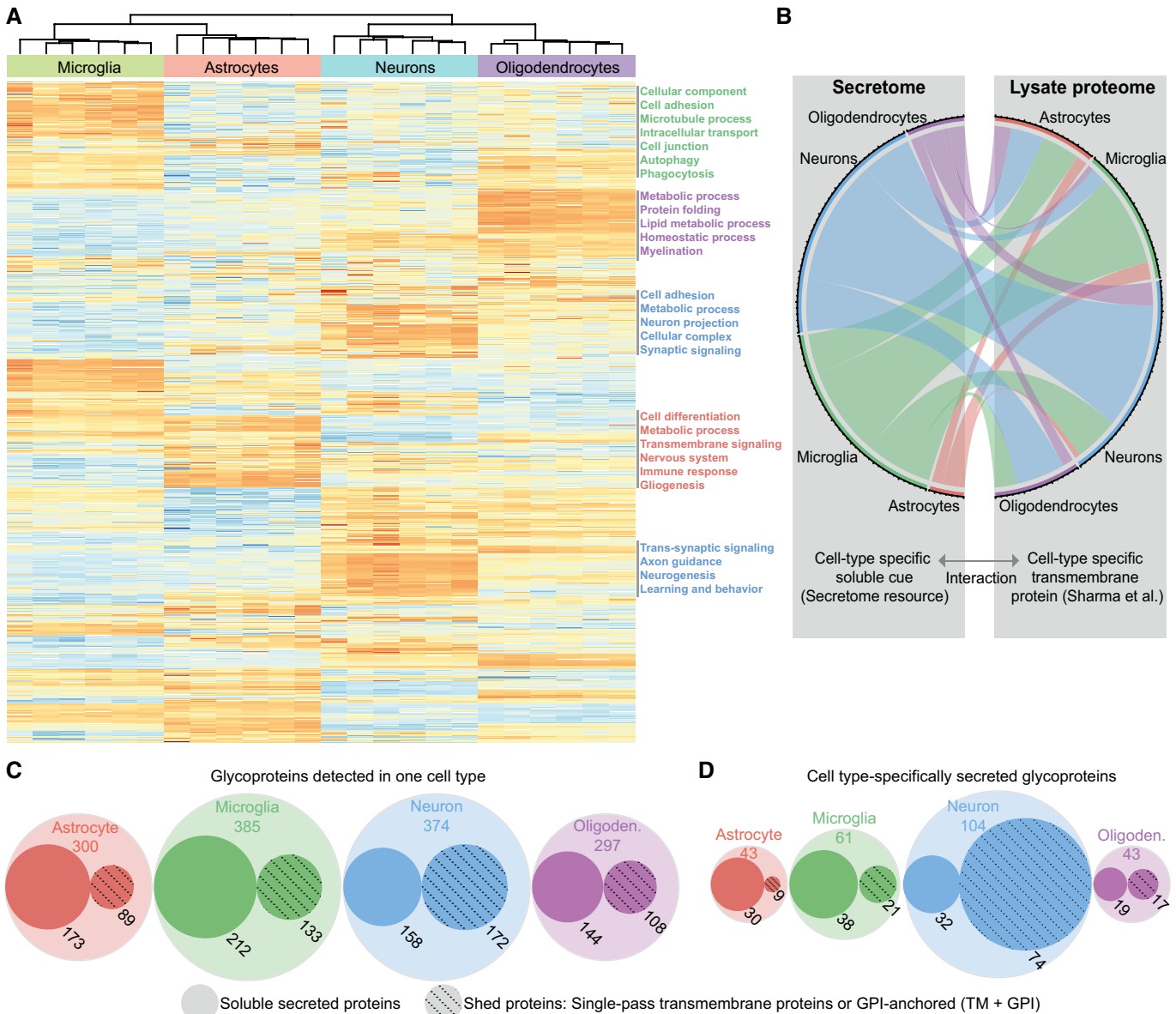

**Figure 3. Cell type-specific enrichment of proteins in the secretome of brain cells.**

A Heat map of the hiSPECS library proteins across the four cell types from hierarchical clustering. For missing protein data, imputation was performed. Rows represent the 995 proteins, and columns represent the cell types with their replicates. The colors follow the z-scores (blue low, white intermediate, red high). Functional annotation clustering with DAVID 6.8 (da Huang *et al*, 2009a; da Huang *et al*, 2009b) for the gene ontology category biological process (FAT) of protein clusters enriched in one cell type is indicated on the right sorted by the enrichment score (background: all proteins detected in the hiSPECS brain secretome resource).

B Interaction map of the hiSPECS secretome and the published lysate proteome data (Sharma *et al*, 2015) illustrating the cellular communication network between the major brain cell types. Interaction pairs are based on binary interaction data downloaded from UniProt and BioGRID databases (Chatr-Aryamontri *et al*, 2015; The UniProt Consortium, 2018). Cell type-specifically secreted proteins of the hiSPECS secretome resource were mapped to their interaction partners if the interaction partners also show cell type specificity in lysates of one brain cell type in the proteome data (Sharma *et al*, 2015) (2.5-fold pairwise) and are annotated as transmembrane protein in UniProt (Table EV5).

C Visualization of the glycoproteins detected in at least 5 of 6 biological replicates of the four brain cell type secretomes. Soluble secreted or potentially shed proteins are indicated for each cell type. The radius of the circles resembles the protein count. Shed proteins include proteins annotated as single-pass transmembrane and glycosylphosphatidylinositol (GPI)-anchored proteins, of which the shed ectodomain was found in the secretome.

D Visualization of the cell type-specific (CTS) glycoproteins as in (C), either specifically secreted by one cell type in at least 5 biological replicates and no more than 2 biological replicates in another cell type or fivefold enriched according to pairwise comparisons to the other cell types (Table EV4).

hippocampal neurons to determine whether identical or different BACE1 substrates would be identified compared with the culture above, that mostly contained cortical, but also hippocampal neurons. The secretome of hippocampal versus cortical neurons correlated very well (*R* = 0.85, Fig EV5D) showing an overlap of 82% of commonly identified, secreted proteins (Fig EV5E). One of

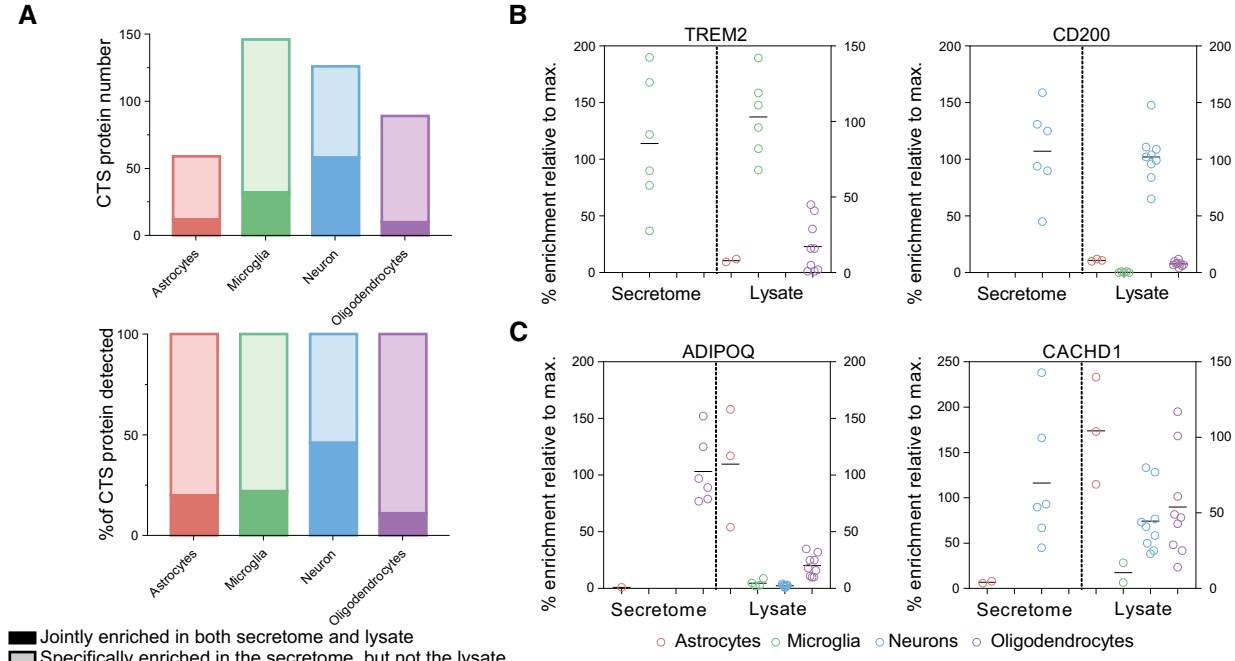

**Figure 4. Protein levels in the brain cell secretome vs. lysate proteome.**

A    Bar graph of proteins specifically secreted from the indicated cell types. The solid part of the box indicates which fraction of proteins were predominantly enriched both in the secretome and in the lysate of the indicated cell type, suggesting that cell type-specific secretion results from cell type-specific protein synthesis. The light part of the box shows the fraction of proteins that were specifically secreted from the indicated cell type, although this protein had similar levels in the lysate of multiple cell types, indicating cell type-specific (CTS) mechanisms of secretion or shedding. Lysate protein levels were extracted from (Sharma et al, 2015).

B, C    Comparison of the hiSPECS secretome resource and lysate data by (Sharma et al, 2015). The % enrichment is indicated normalized to the average of the most abundant cell type. (B) TREM2 and CD200 are jointly enriched in both secretome and lysate in microglia or neurons, respectively. In (C), two examples, of proteins specifically enriched in the secretome, but not in the lysate, are shown. ADIPOQ is specifically secreted from oligodendrocytes, but reveals highest expression in astrocytes. CACHD1 is specifically secreted from neurons, but high protein levels can be found in astrocytes, neurons, and oligodendrocytes. The black line indicates the mean.

the proteins only identified in the hippocampal secretome was TMEM108, in agreement with its high expression in the hippocampus (Sharma et al, 2015). TMEM108 is involved in adult hippocampal neurogenesis and linked to diverse psychiatric disorders (Yu et al, 2019). Next, we determined how the BACE1 inhibitor C3 altered the secretome of hippocampal neurons (Fig EV5F and G) and identified largely the same BACE1 substrate candidates as in the mixed, mostly cortical neuron culture (Fig EV5A), suggesting that the difference in BACE1 expression between cortex and hippocampus does not have a major effect on BACE1 substrate cleavage and identification.

The proteolytically inactive ADAM22, which is a new BACE1 substrate candidate, and CD200, which was previously suggested as a BACE1 substrate candidate in a peripheral cell line (Stutzer et al, 2013), were further validated as neuronal BACE1 substrates by Western blots and ELISAs (Fig 5C–E). For CD200, we also detected a semi-tryptic peptide in the conditioned medium of neurons and in a previous proteomic dataset (Pigoni et al, 2016) of murine cerebrospinal fluid (CSF; Fig 5F and G), but not when BACE1 was inhibited in the neurons or in the CSF of mice lacking BACE1 and its homolog BACE2 (Pigoni et al, 2016). As this semi-tryptic peptide derives from the juxtamembrane domain of CD200 where BACE1 typically cleaves its substrates, this peptide likely represents the cleavage site of BACE1 in CD200 (Fig 5G). The validation of CD200

as a new in vivo BACE1 substrate demonstrates the power of hiSPECS to unravel the substrate repertoire of transmembrane proteases and reveals that cell type-specific expression of an ectodomain shedding protease is an important mechanism controlling the cell type-specific secretion/shedding of proteins.

Taken together, cell type-specific protein secretion and shedding are minimally dependent on cell type-specific protein expression of the secreted protein. Instead, cell type-specific expression of shedding regulators and other mechanisms to be discovered have an important role in determining cell type-specific protein secretion.

### Cell type-specific origin of CSF proteins

Cerebrospinal fluid (CSF) is the body fluid in direct contact with the brain and serves as an in vivo secretome. It is widely used for basic and preclinical research as well as clinical applications. However, a major limitation of CSF studies is that changes in CSF composition induced by disease or treatments often cannot be traced back to the cell type of origin, because most CSF proteins are expressed in multiple cell types (Sharma et al, 2015). To overcome this limitation, we applied the cell type-resolved brain secretome resource derived from the four most abundant brain cell types to determine the likely cellular origin of secreted glycoproteins in CSF (Table EV7).

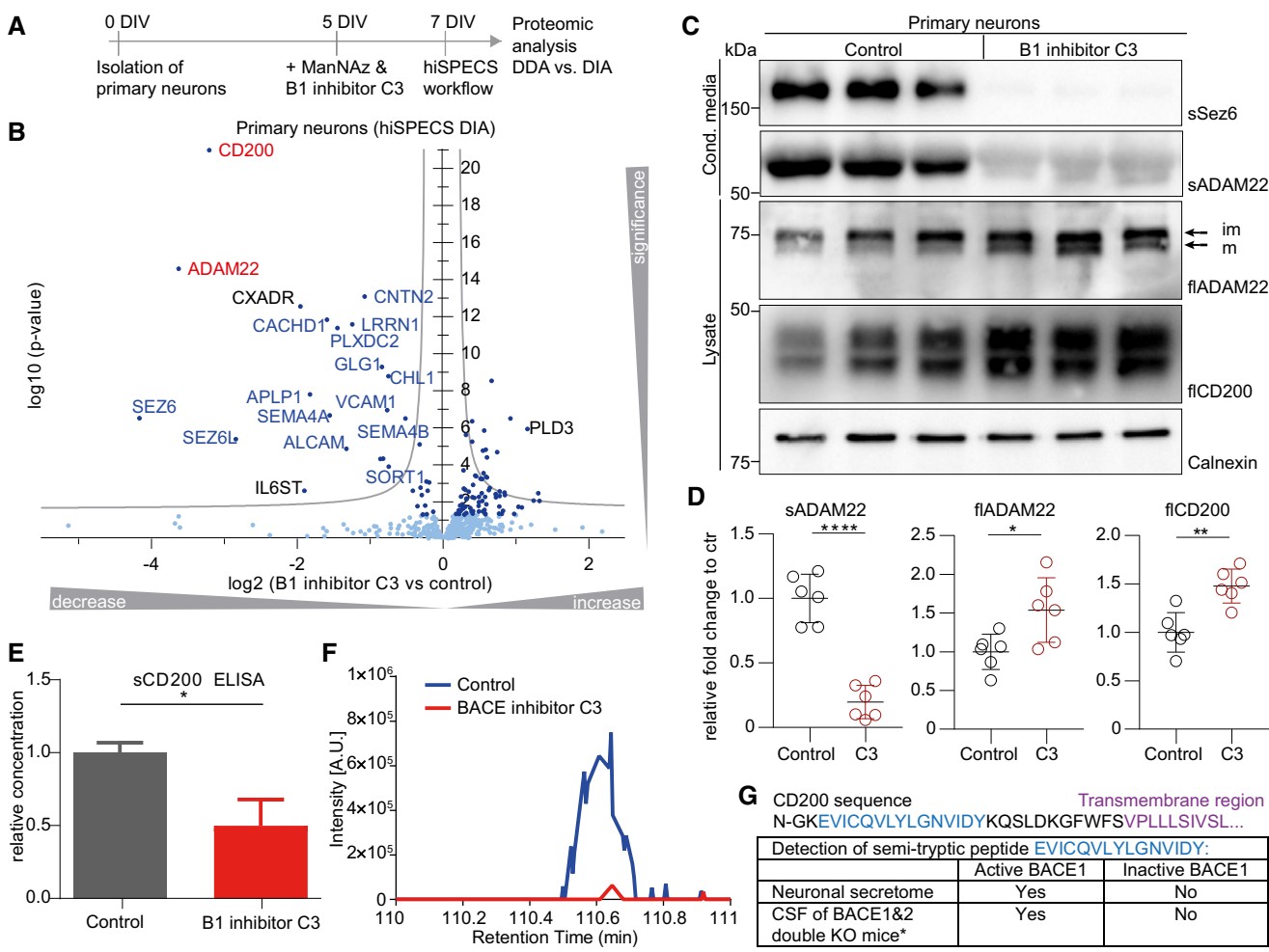

**Figure 5. Identification and validation of substrate candidates of the protease BACE1.**

A Experimental design of the ManNAz labeling step and BACE1 inhibitor C3 treatment of the primary neuronal culture for 48 h at 5–7 days *in vitro* (DIV).

B Volcano plot showing changes in protein levels in the secretome of primary neurons upon BACE1 inhibitor C3 treatment using the hiSPECS DIA method. The negative $\log_{10}$ *P*-values (two-sample *t*-test) of all proteins are plotted against their $\log_2$ fold changes (C3 vs control) (*N* = 11). The gray hyperbolic curves depict a permutation-based false discovery rate estimation (*P* = 0.05; s0 = 0.1). Significantly regulated proteins (*P* < 0.05) are indicated with a dark blue dot, and known BACE1 substrates are indicated with blue letters. The two newly validated BACE1 substrates CD200 and ADAM22 are indicated in red.

C Independent validation of the novel BACE1 substrate candidates CD200 and ADAM22 by Western blotting in supernatants and lysates of primary neurons incubated with or without the BACE1 (B1) inhibitor C3 for 48 h. Full-length (fl) ADAM22 (mainly mature ADAM22, lower band) and CD200 levels in the neuronal lysate were mildly increased upon BACE1 inhibition, as expected due to reduced cleavage by BACE1. Calnexin served as a loading control. The soluble ectodomain of ADAM22 (sADAM22) was strongly reduced in the conditioned medium upon BACE1 inhibition. Ectodomain levels of the known BACE1 substrate SEZ6 (sSEZ6) were strongly reduced upon BACE1 inhibition and served as positive control. Arrows indicate the mature (m) and immature (im) (before prodomain cleavage) form of ADAM22.

D Quantification of the Western blots in (C) (*N* = 6). Signals were normalized to calnexin levels and quantified relative to the control (ctr) condition. Statistical testing was performed with *N* = 6 biological replicates, using the one-sample *t*-test with the significance criteria of *P* < 0.05. According to this criterion, ADAM22 and CD200 were significantly increased in total lysates upon C3 treatment (flADAM22: *\**P*-value 0.0251, flCD200: **\**P*-value 0.011). Soluble ADAM22 was significantly reduced in the supernatant upon C3 treatment (****\**P*-value < 0.0001). The black central horizontal line indicates the mean and error bars, mean ± SD.

E The reduction in the soluble ectodomain of CD200 (sCD200) was detected by ELISA (one-sample *t*-test, *\**P*-value 0.0116), because the available antibodies were not sensitive enough for Western blots of the conditioned medium (*N* = 4). The mean and error bars are shown, mean ± SD.

F Extracted ion chromatogram of the semi-tryptic peptide of CD200 in conditioned media of neurons comparing C3-treated to control condition. Levels of the semi-tryptic peptide were strongly reduced upon BACE1 inhibition.

G The potential cleavage site of CD200 was identified by a semi-tryptic peptide indicated in blue, which is from the juxtamembrane domain of CD200. The transmembrane domain is indicated in purple. The semi-tryptic peptide was found (i) using the hiSPECS method in the neuronal secretome under control conditions but not upon BACE1 inhibition, and ii) in the CSF of wild-type mice but not upon knockout of BACE1 and its homolog BACE2—*data are extracted from (Pigoni *et al*, 2016). Because BACE2 is hardly expressed in brain, both datasets demonstrate that generation of the semi-tryptic peptide requires BACE1 activity and thus represents the likely BACE1 cleavage site in CD200.

Source data are available online for this figure.

In CSF from individual wild-type mice, 984 protein groups were identified with DIA in at least 3 out of 4 biological replicates (Fig 6A), which represents higher coverage than in previous mouse CSF studies (Dislich *et al*, 2015; Pigoni *et al*, 2016) and underlines the superiority of DIA over DDA in samples with low protein concentration. Proteins were grouped according to their $\log_{10}$ DIA

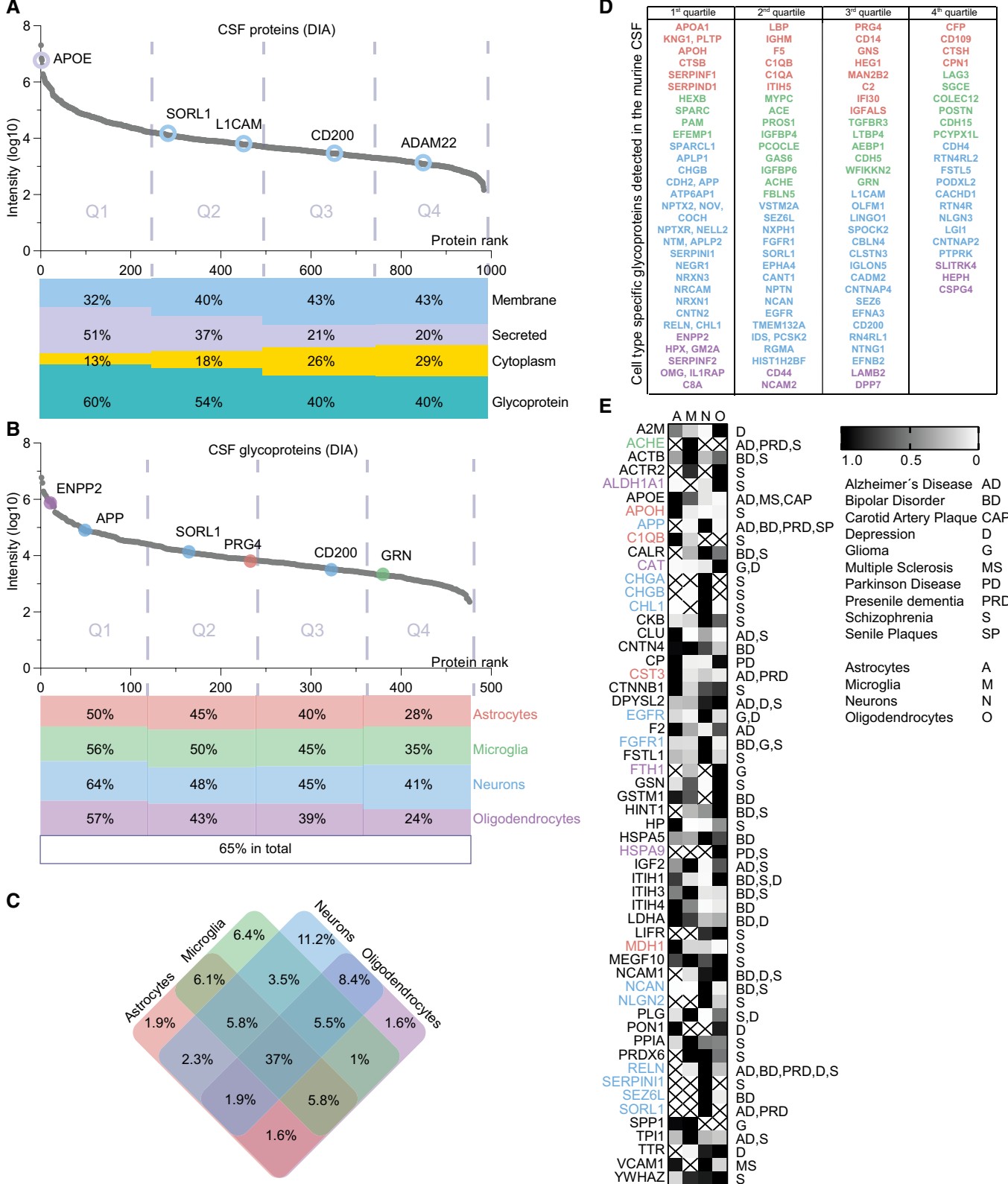

Figure 6.

**Figure 6.  Mapping of murine CSF proteins to their probable cell type origin.**

A    Protein dynamic range plot of the $\log_{10}$-transformed LFQ intensities of the murine CSF proteins quantified in at least 3 of 4 biological replicates measured with DIA. The proteins are split into quartiles according to their intensities, with the 1st quartile representing the 25% most abundant proteins. The percentage of proteins annotated in UniProt with the following subcellular locations/keywords are visualized for membrane, secreted, cytoplasm, and glycoprotein. Examples of cell type-specifically secreted proteins are indicated with circles colored according to the UniProt annotation (purple: secreted, blue: membrane).

B    Protein dynamic range plot of the $\log_{10}$-transformed LFQ intensities specifically of glycoproteins in the murine CSF quantified in at least 3 of 4 biological replicates measured with DIA ($N = 4$). The proteins are split into quartiles according to their intensities. The percentage of proteins identified in the secretome of astrocytes, microglia, neurons, or oligodendrocytes in at least 5 of 6 biological replicates is illustrated below. Selected proteins specifically secreted from one cell type are indicated with the color code of the corresponding cell type.

C    Venn diagram indicating the distribution of CSF glycoproteins detected in the hiSPECS secretome resource.

D    Cell type-specifically secreted proteins (CTSP) in the hiSPECS secretome study (fivefold enriched in pairwise comparison with the other cell types or only detected in one cell type) are listed according to their presence in the CSF quartiles. CTSP are color-coded according to their origin: astrocytes (orange), microglia (green), neurons (blue), and oligodendrocytes (purple).

E    List of proteins detected in murine CSF and the hiSPECS secretome resource which have human homologs that are linked to brain disease based on the DisGeNET database (Pinero *et al*, 2017) with an experimental index, e.i ≥ 0.9. Relative protein levels in the brain cell secretome are indicated (black high, white low abundance). Colored gene names indicate cell type-specific secretion. Columns indicate astrocytes (A), microglia (M), neurons (N), and oligodendrocytes (O). X: not detected in secretome.

LFQ intensities (as a rough estimate of protein abundance) into quartiles and analyzed according to their UniProt annotations for membrane, secreted, and cytoplasm (Fig 6A). Soluble secreted proteins, such as APOE, were more abundant in the 1st quartile, whereas the shed ectodomains, e.g., of CD200 and ADAM22, were the largest group of proteins in the 3rd and 4th quartile, indicating their lower abundance in CSF as compared to the soluble secreted proteins.

Among the 476 CSF proteins annotated as glycoproteins (UniProt), 311 (65%) were also detected in the hiSPECS secretome resource and were mapped to the corresponding cell type (Fig 6B). The most abundant glycoproteins of each cell type (top 25) revealed also high coverage in the CSF up to 76% of the top 25 neuronal proteins (Appendix Fig S2). In general, proteins of neuronal origin represented the largest class of CSF glycoproteins and this was independent of their abundance (Fig 6B and C). Given that astrocytes and oligodendrocytes outnumber neurons by far in the brain, this demonstrates that neurons disproportionately contribute to the CSF proteome. This prominent role of neurons is also reflected when focusing on the 420 cell type-specifically secreted proteins of our brain secretome resource. 123 of these proteins were also found in murine CSF (Fig 6D). The majority in each quartile were proteins of neuronal origin. Similar to the proteins secreted from the four major brain cell types *in vitro*, the largest amount of the cell type-specifically secreted proteins detected in CSF is expressed in multiple cell types, but only secreted from one (Table EV4). This includes BACE1 substrates, such as SEZ6L and CACHD1, in agreement with BACE1 being expressed in neurons, but not in other brain cell types (Voytyuk *et al*, 2018). Thus, cell type-specific protein expression as well as cell type-specific protease expression and potentially additional mechanisms governs cell type-specific protein secretion, both *in vitro* (hiSPECS resource) and *in vivo* (CSF).

Several of the detected murine CSF proteins have human homologs linked to brain diseases (DisGeNET database; Pinero *et al*, 2017) and may serve as potential biomarkers. We mapped these proteins to their likely cell type of origin (Fig 6E). Numerous proteins were specifically secreted from only one cell type, such as APP from neurons or granulin (GRN) from microglia, which have major roles in neurodegenerative diseases (O'Brien & Wong, 2011; Chitramuthu *et al*, 2017). Thus, assigning the disease-related CSF/secretome proteins to one specific cell type

offers an excellent opportunity to study the relevant cell type with regard to its contribution to disease pathogenesis. One example is the protein SORL1, which is genetically linked to Alzheimer's disease (Yin *et al*, 2015). Although SORL1 is similarly expressed in all four major brain cell types, it was specifically released from neurons in the hiSPECS resource (Table EV4), indicating that pathology-linked changes in CSF SORL1 levels are likely to predominantly result from neurons.

Large-scale proteomic analyses of AD brain tissue and CSF (Bai *et al*, 2020; Johnson *et al*, 2020) continue to reveal more AD-linked proteins, such as increased CD44 in the CSF of AD patients (Johnson *et al*, 2020). Our resource demonstrates that CD44 is predominantly secreted from oligodendrocytes among the mouse brain cells, although having similar protein abundance in different brain cell types (Sharma *et al*, 2015) (Fig EV4). These data suggest to focus on oligodendrocytes for future studies determining how increased CD44 levels are mechanistically linked to AD pathophysiology. Taken together, the hiSPECS resource enables systematic assignment of CSF glycoproteins to the specific cell type of origin, which offers multiple opportunities to study CNS diseases.

**Cell type-resolved brain tissue-secretome**

Next, we tested whether hiSPECS and the brain secretome resource can be used to determine the cell type-resolved secretome of brain tissue. We used organotypic cortico-hippocampal brain slices (Table EV8), an *ex vivo* model of the brain (Daria *et al*, 2017) that preserves the complex network of the diverse brain cell types. Despite the high amounts (25%) of serum proteins, 249 protein groups were identified in at least 5 of 6 biological replicates using hiSPECS DIA (Fig 7A), demonstrating that hiSPECS is applicable for *ex vivo* brain tissue. Proteins were grouped according to their $\log_{10}$ abundance into quartiles (Fig 7B). Similar to CSF, the majority of the more abundant proteins in the 1st and 2nd quartile were soluble proteins, whereas the shed transmembrane protein ectodomains were in the 3rd and 4th quartile, indicating their lower protein abundance compared with the soluble secreted proteins. 89% of the proteins detected in the slice culture secretome were also detected in the hiSPECS secretome library of the different brain cell types, which allows tracing them back to their cellular origin (Fig 7B). The cell type-resolved tissue secretome revealed on average the highest

contribution of microglia (74%), followed by astrocytes (72.8%), oligodendrocytes (68.5%), and neurons (66.5%). Interestingly, in quartile 1, which resembles the most abundant proteins, oligodendrocytes are the most prominent with 85%. In addition, numerous cell type-specifically secreted proteins were identified (Fig 7C).

As a final application of the brain secretome resource, we treated brain slices for 6 h with the strong inflammatory stimulus

lipopolysaccharide (LPS), which serves as a model for acute neuroinflammation. Conditioned medium was analyzed using hiSPECS DIA. LPS strongly changed the brain slice secretome. Secretion of several proteins (marked in red) known to be LPS-responsive in macrophages (Meissner *et al*, 2013), such as H2-D1, CD14, or IL-12B, was strongly upregulated (up to 250-fold; Fig 7D, Table EV8). Likewise, secretion of several proteins not known to be secreted in

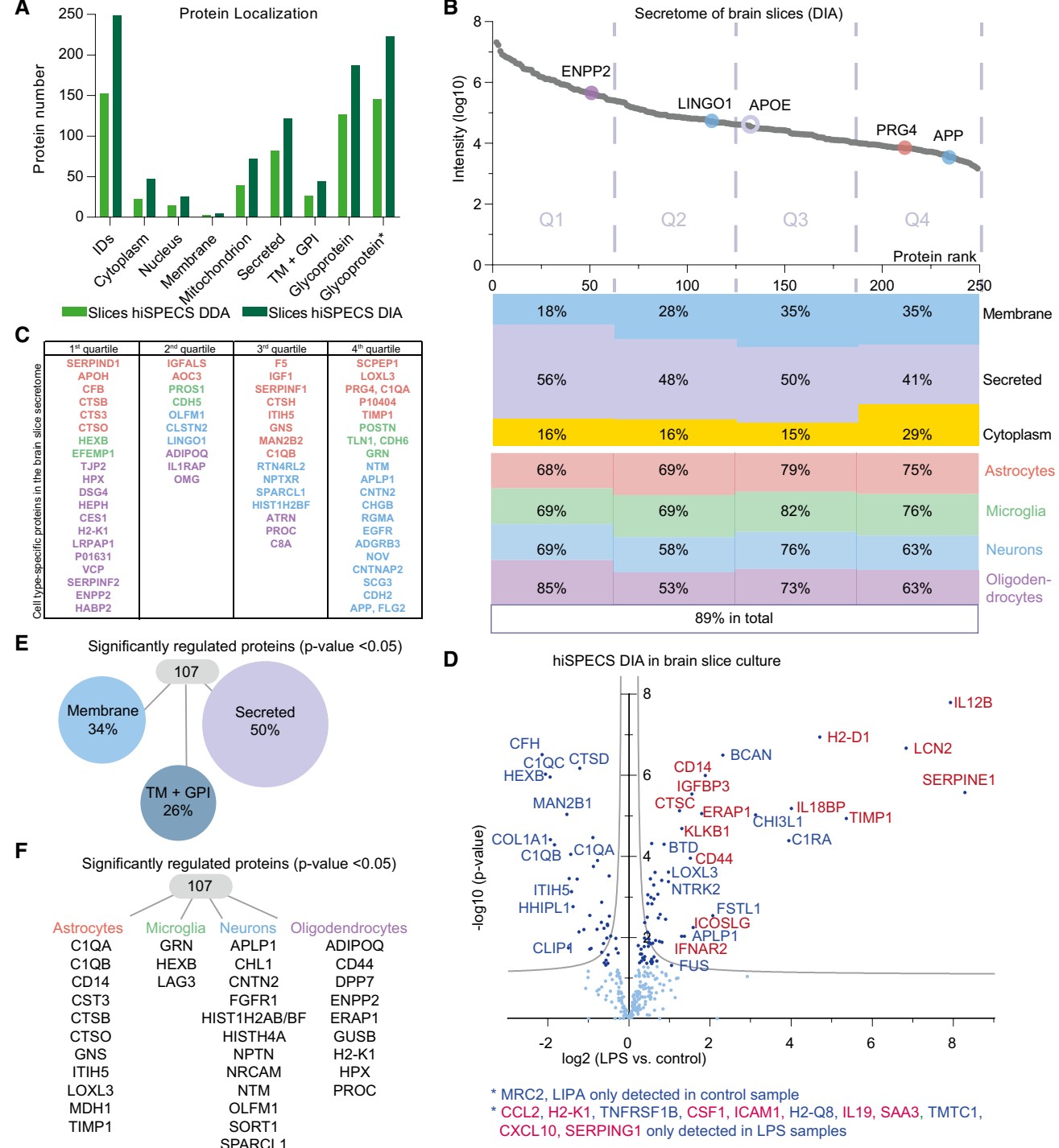

Figure 7.

**Figure 7. The secretome of brain slices.**

A  hiSPECS DDA and DIA analysis of brain slice cultures in the presence of 25% serum. The bar chart comparing the hiSPECS DDA and DIA method indicates the protein number and their localization according to UniProt identified in the secretome of brain slices. Proteins quantified in at least 5 of 6 biological replicates are considered (*N* = 6).

B  Protein dynamic range plot of the $\log_{10}$ LFQ intensities of secretome proteins of brain slices in descending order. According to their intensity, proteins are grouped into quartiles and the percentage of proteins with the following UniProt keywords is visualized: membrane, secreted, and cytoplasm. The percentage of proteins identified in the secretome of astrocytes, microglia, neurons, or oligodendrocytes in at least 5 of 6 biological replicates is illustrated. Selected proteins specifically secreted from one cell type are indicated with the color code of the corresponding cell type. The circle of APOE indicates it as a soluble secreted protein.

C  Cell type-specifically secreted proteins (CTSP) according to the hiSPECS secretome resource (fivefold enriched in pairwise comparison with the other cell types or consistently detected only in one cell type; Table EV4) detected in the secretome of brain slices. CTSP are color-coded according to their origin: astrocytes (orange), microglia (green), neurons (blue), and oligodendrocytes (purple).

D  Volcano plot showing changes in protein levels in the secretome of primary cultured brain slices upon 6 h LPS treatment in a 24 h collection window using the hiSPECS DIA method. The negative $\log_{10}$ *P*-values (two-sample *t*-test) of all proteins are plotted against their $\log_2$ fold changes (LPS vs control) (*N* = 6). The gray hyperbolic curves depict a permutation-based false discovery rate estimation (*P* = 0.05; s0 = 0.1). Significantly regulated proteins (*P* < 0.05) are indicated with a dark blue dot. Proteins highlighted in red indicate proteins known to increase upon LPS treatment, whereas proteins labeled in blue are not upregulated in this study (Meissner *et al*, 2013); for details, see Table EV8.

E, F  Significantly regulated proteins upon LPS treatment (*P* < 0.05, two-sample *t*-test) of brain slices split according to UniProt keywords membrane, single-pass transmembrane and GPI-anchored, and secreted proteins; (F) indicating cell type specificity in the hiSPECS secretome resource.

an LPS-dependent manner, such as BCAN, CHI3L1, and the complement receptor C1RA, was strongly upregulated. Among the 107 significantly regulated proteins (*P* < 0.05), 50% and 26% are annotated as secreted proteins or single-pass transmembrane proteins, respectively (Fig 7E). Comparison of the brain slice secretome data to the secretome resource revealed that several proteins were secreted cell type-specifically, such as SORT1 and CHL1 by neurons and ENPP5 and ADIPOQ by oligodendrocytes (Fig 7F). This demonstrates that not only immune cells responded with changes in their secretome to the inflammation cue, but instead indicates a systemic inflammatory response of multiple cell types. Moreover, this experiment suggests that systematic, proteome-wide secretome analysis of *ex vivo* brain slices is well-suited to identify proteins and cell types contributing to neuroinflammation and potentially neurodegeneration.

## Discussion

Omics' approaches have generated large collections of mRNA and protein abundance data across different cell types, including microglia and neurons, but we know very little about the molecules that are secreted from cells. This information is essential for our understanding of basic mechanisms of protein secretion, cell–cell communication within organs, particularly within the brain, and for identification of suitable biomarkers for brain processes in health and disease, such as APOE and TREM2 in AD (Wolfe *et al*, 2018; Huang *et al*, 2019).

Our development of the novel method hiSPECS miniaturizes mass spectrometry-based secretome analysis and enables secretome analysis of primary cell types, including the lower abundant ones in the brain. Importantly, hiSPECS allows culturing cells in the presence of serum or physiological cell culture supplements, whereas most previous secretome studies were restricted to serum-free and even protein-free culture conditions, which is not feasible with many primary cell types. Besides its broad applicability to primary cells, we demonstrate that hiSPECS can also be applied to brain tissue *ex vivo*, which is widely used in neuroscience. The streamlined hiSPECS workflow also facilitates a broad applicability in laboratories without proteomic expertise. The strongly reduced mass

spectrometer measurement time enables cost-effective, single-shot proteomic analysis of the samples.

With hiSPECS, we established the cell type-resolved secretome resource of the four major cell types in the brain and its application to map the putative cell type-specific origin of CSF proteins (*in vivo* secretome) and proteins secreted from brain slices (*ex vivo* secretome). This approach provided fundamental new biological insights. First, ectodomain shedding is a major mechanism contributing to the protein composition of the secretome and quantitatively differs between brain cell types. Second, shed proteins *in vitro* (primary cells), *ex vivo* (brain slices), and *in vivo* (CSF) have a generally lower abundance in the secretome than secreted soluble proteins. This is consistent with the function of the shedding process as a regulatory mechanism, which releases bioactive membrane protein ectodomains on demand into the secretome, as seen with cytokines (Lichtenthaler *et al*, 2018). Thus, shedding provides an additional layer of control for the composition of the secretome that goes beyond constitutive protein secretion. Third, protein secretion is a highly cell type-specific process in the nervous system. This is surprising because we found that more than 73% of the cell type-specifically secreted proteins were expressed in more than one cell type, demonstrating that cells do not simply control secretion through cell type-specific expression of the secreted protein, but instead must have acquired additional mechanisms to control cell type-specific protein secretion, which are not yet well-understood. Our resource provides insights into the underlying mechanisms. One example is the cell type-specific expression of a shedding protease, such as BACE1 in neurons. Ectodomain shedding happens for more than 1,000 membrane proteins (Lichtenthaler *et al*, 2018), but in most cases, the contributing protease is not known. Thus, it is likely that shedding proteases other than BACE1 are also expressed in a cell type-specific manner and thus contribute to cell type-specific protein shedding in the brain.

While additional mechanisms underlying cell type-specific protein secretion remain to be elucidated, protein transport through the secretory pathway, which is a prerequisite for protein secretion or shedding, is a potential mechanism of regulation. In fact, some soluble proteins require CAB45 for their exit from the trans-Golgi network, whereas other proteins do not (Blank & von Blume, 2017). Likewise, some transmembrane proteins require

specific transport helpers for trafficking through the secretory pathway such as IRHOM1/2 for ADAM17 (Adrain *et al*, 2012; McIlwain *et al*, 2012) and Cornichon for transforming growth factor (Dancourt & Barlowe, 2010). Thus, it appears possible that transport-selective proteins may be preferentially expressed in some brain cell types and consequently allow for a cell type-specific secretion or shedding of their cargo proteins. hiSPECS is an excellent method to address these fascinating questions regarding the complex mechanisms controlling protein secretion in the nervous system.

The secretome of the four major brain cell types and the *ex vivo* tissue identified here represents a snapshot of the total brain secretome, and more secreted proteins are known or likely to exist. These include non-glycosylated and non-sialylated secreted proteins that are not captured with hiSPECS as well as proteins secreted in a time-dependent manner such as during development or aging. For example, it is known that microglia can partially change their expression profile when taken into the culture, which may consequently affect the secretome (Gosselin *et al*, 2017). Additionally, the secretome may change upon cell stimulation, such as during neuronal activity or inflammation or when different cell types are cocultured or taken into three-dimensional culture systems (Stiess *et al*, 2015). Additional proteins may be selectively secreted from lower abundant brain cell types, such as pericytes.

Taken together, hiSPECS and the cell type-resolved mouse brain secretome resource are important new tools for many areas in neuroscience, from mechanisms of protein secretion and signal transduction between brain cells *in vitro*, *ex vivo* (brain slices), and *in vivo* (CSF) to functional analysis of nervous system proteins (identification of protease substrates) and cell type-specific biomarker determination in CSF (e.g., CD44) with high relevance to psychiatric, neurological, and neurodegenerative diseases.

# Materials and Methods

## Reagents and Tools table

| Reagent/Resource | Reference or Source | Identifier or Catalog Number |
|---|---|---|
| **Experimental Models** | | |
| Primary cultured astrocytes, microglia, neurons, oligodendrocytes of C57BL/6J wildtype mice (*M. musculus*) | The Jackson Laboratory/Charles River | RRID:IMSR_JAX:000664 |
| **Antibodies** | | |
| Goat polyclonal anti-CD200 | R and D systems | Cat# AF2724, RRID: AB_416669 |
| Mouse monoclonal anti-ADAM22 | UC Davis/NIH NeuroMab Facility | Cat#75-093; RRID: AB_2223817 |
| Rat monocolonal anti-SEZ6 | Pigoni *et al* (2016) | N/A |
| Rabbit polyclonal anti-calnexin | Enzo Life Sciences | Cat#ADI-SPA-860, RRID: AB_10616095 |
| **Chemicals, Enzymes and other reagents** | | |
| Mouse CD200 (Sandwich ELISA) ELISA Kit | LSBio | Cat#LS-F2868 |
| Neural Tissue Dissociation Kit (P) | Miltenyi Biotec | Cat#130-092-628 |
| CD11b (Microglia) MicroBeads, human and mouse | Miltenyi Biotec | Cat# 130-093-634 |
| Anti-AN2 MicroBeads, human and mouse | Miltenyi Biotec | Cat#130-097-170 |
| β-Secretase Inhibitor IV − C3, Calbiochem | Sigma Aldrich | Cat#565788 |
| tetra-acetylated N-azidomannosamine (ManNAz) | Thermo Fisher Scientific | Cat#C33366 |
| magnetic DBCO beads | Jena Bioscience | Cat#CLK-1037-1 |
| Concanavalin A agarose conjugate | Sigma Aldrich | Cat#C7555 |
| Sodium deoxycholate | Sigma Aldrich | Cat#30970 |
| Ammonium bicarbonate | Sigma Aldrich | Cat#5.33005 |
| Trypsin | Promega | Cat#V5111 |
| LysC | Promega | Cat#V1671 |
| ReproSil-Pur 120 C18-AQ, 1.9 µm | Dr. Maisch GmbH | Cat#r119.aq. |
| 30 cm Uncoated SilicaTip Emitters | New Objective | Cat#FS360-75-8-N-5-C30 |
| **Software** | | |
| MaxQuant (version 1.5.5.1 or 1.6.1.0) | Jürgen Cox, Max-Planck-Institute of Biochemistry | |

**Reagents and Tools table** (continued)

| Reagent/Resource | Reference or Source | Identifier or Catalog Number |
|---|---|---|
| | | RRID:SCR_014485 https://maxquant.net/maxquant/ |
| Perseus (Version 1.6.6.0) | Jürgen Cox, Max-Planck-Institute of Biochemistry | SCR_015753 https://maxquant.net/perseus/ |
| R | The R Fundation for Statistical Computing | https://www.r-project.org/ |
| Spectronaut™ Pulsar X | Biognosys | N/A |
| **Other** | | |
| Q Exactive™ HF Hybrid Quadrupol-Orbitrap™ mass spectrometer | Thermo Fisher Scientific | IQLAAEGAAPFALGMBFZ |
| EASY-nLC 1200 | Thermo Fisher Scientific | Cat#LC140 |

## Methods and Protocols

### Mice

All murine samples were isolated from C57BL/6J mice from the Jackson Laboratory according to the European Communities Council Directive. Mice were housed and breed in the pathogen-free animal facility of the DZNE Munich.

### Primary cell culture and brain slices

All primary cultures were maintained under standard cell culture conditions at 37°C with 5% $CO_2$. The samples were collected from at least two independent culture preparations, with 3 dishes of primary cells from each culture preparation. It is not possible to determine the sex of the cells, because the cells were isolated from embryos or young pups. The conditioned media were stored at −20°C until further processing.

Primary neurons were isolated at E16.5 as described before (Kuhn *et al*, 2012). Meninges-free cerebral cortices and/or hippocampi were dissociated and digested in DMEM with 200 U Papain for 30 min (Sigma-Aldrich) and plated into poly-D-lysine-coated 6 wells in plaiting media (DMEM + 10% FBS + 1% penicillin/streptomycin). After 4 h, media were changed to neuronal cultivation media (B27 + neurobasal + 0.5 mM glutamine + 1% P/S). Mixed hippocampal and cortical neuronal cultures were used for the benchmarking experiment in Fig 1, and BACE1 inhibitor experiment and the validation of CD200 and ADAM22 shown in Figs 5 and EV5A. Pure cortical neurons were used for the comparison of hippocampal and cortical neurons shown in Fig EV5C. Pure hippocampal neurons were used for the comparison of the four brain cell types in the secretome resource (e.g., Fig 2), the comparison between hippocampal and cortical neurons (Fig EV5C) the BACE1 inhibitor experiment (Fig EV5F). Of note, the proteomic raw files of the hiSPECS hippocampal neuronal secretome analysis ($N = 6$) were used twice in this project, first for the comparison of the four major brain cell types (Table EV2) and secondly for the comparison of hippocampal to cortical neurons (Table EV6).

Primary astrocytes were isolated at E16.5 and dissociated in the same way as the primary neurons; however, they were plated on uncoated dishes. Cultures were grown until reaching confluence in DMEM + 10% FBS, and cells were detached using trypsin and re-seeded on a new plate (50% confluence). This procedure was repeated three times before seeding the cells for the final experiment ($1 \times 10^6$ cells into a well of a 6-well plate).

Primary oligodendrocyte progenitor cell (OPC) cultures were prepared by magnetic-activated cell sorting (MACS). 60-mm cell culture dishes were coated with 0.01% poly-L-lysine for 1 h at 37°C, washed twice, and incubated with MACS Neuro Medium (Miltenyi Biotec, #130-093-570) overnight. OPCs were isolated from the brains of postnatal day 6 C57BL/6J mouse pups. Cell suspension was obtained by automated dissociation using the Neural Tissue Dissociation Kit (P) (Miltenyi Biotec, Cat #130-092-628) and the gentle-MACS™ Dissociator (Miltenyi Biotec, Cat#130-093-235) following the datasheet of the kit with some modifications. DMEM/pyruvate medium was used instead of HBSS during tissue dissociation. All media were warmed up to room temperature. The optional centrifugation steps were included in the dissociation. The myelin removal step was omitted. Prior to labeling with anti-AN2 MicroBeads (Miltenyi Biotec, Cat#130-097-170), the cell suspension was incubated with the OPC MACS cultivation medium (MACS Neuro Medium containing MACS NeuroBrew-21 (Miltenyi Biotec, #130-093-566), GlutaMAX (Thermo Fisher Scientific, #35050087) and penicillin/streptomycin) for 3 h at 37°C for surface antigen re-expression. DMEM containing 1% horse serum and penicillin/streptomycin (DMEM/HS medium) was used as the buffer for magnetic labeling and separation. After magnetic separation, the OPC MACS cultivation medium was applied to flush out AN2$^+$ cells. $1 \times 10^6$ cells were plated in 4 mL of OPC MACS cultivation medium per 60-mm dish. For OPC/oligodendrocyte culture, ManNAz was added directly to the medium after one day *in vitro* and incubated for 48 h.

Primary microglia were isolated from postnatal day 5 mouse brains using the MACS technology as previously described (Daria *et al*, 2017). Briefly, olfactory bulb, brain stem, and cerebellum were removed and the remaining cerebrum was freed from meninges and dissociated by enzymatic digestion using the Neural Kit P (Miltenyi Biotec; Cat #130-092-628). Subsequently, tissue was mechanically dissociated by using three fire-polished glass Pasteur pipettes of decreasing diameter. Microglia were magnetically labeled using CD11b MicroBeads (Miltenyi Biotec, Cat#130-093-634), and cell suspension was loaded onto a MACS LS Column (Miltenyi Biotec) and subjected to magnetic separation. $1.5–2 \times 10^6$ microglia were

then cultured in DMEM/F12 media (Invitrogen) supplemented with 10% heat-inactivated FBS (Sigma) and 1% penicillin–streptomycin (Invitrogen) in a 60-mm dish for four days before the 48-h treatment with ManNAz. Conditioned media of $1 \times 10^6$ cells were used for the hiSPECS experiment.

Organotypic brain slice cultures from young (postnatal days 6–7) mice were prepared as described previously (Daria *et al*, 2017). Briefly, after brain isolation, olfactory bulb, midbrain, brain stem, and cerebellum were removed and the two remaining cortical hemispheres cut at 350 μm with a tissue chopper (McIlwain, Model TC752, Mickle Laboratory Engineering Company). Intact sagittal cortico-hippocampal slices were selected and incubated for 30 min at 4°C in a pre-cooled dissection media (50% HEPES-buffered MEM, 1% penicillin–streptomycin, 10 mM Tris, pH 7.2). Four slices were then plated onto a 0.4-μm porous polytetrafluoroethylene (PTFE) membrane insert (PICMORG50, Millipore) placed in a 35-mm dish with a slice culture media containing 50% HEPES-buffered MEM, 25% HBSS, 1 mM L-glutamine (Gibco), and 25% heat-inactivated horse serum (Merck-Sigma). Media were exchanged 1 day after preparation and subsequently every 3 days. Brain slices were cultured for 14 days before the 48-h treatment with ManNAz.

### hiSPECS

After washing the primary cells with 1× PBS, cell type-specific growth media containing serum supplements with 50 μM of ManNAz (Thermo Fisher Scientific, Cat #C33366) were added for 48 h. Afterward, conditioned media were collected and filtered through Spin-X 0.45 μm cellulose acetate centrifuge tube filter (#8163, Costar) and stored at −20°C in protein LoBind tubes (Eppendorf) until further usage. Glycoprotein enrichment was performed using 60 μl concanavalin A (ConA) bead slurry per sample (Cat #C7555, Sigma-Aldrich). ConA beads were washed twice with 1 ml of binding buffer (5 mM $MgCl_2$, 5 mM $MnCl_2$, 5 mM $CaCl_2$, 0.5 M NaCl, in 20 mM Tris–HCl pH 7.5) before use. The conditioned medium was incubated with the ConA beads for 2 h in an overhead rotator at 4°C. The ConA beads were pelleted by centrifugation (2,000 *g*, 1 min), and the supernatant containing unbound proteins was discarded. The beads were washed three times with 1 ml binding buffer before adding 500 μl of elution buffer (500 mM methyl-alpha-D-mannopyranoside, 10 mM EDTA in 20 mM Tris–HCl pH 7.5) and rotating overhead for 30 min at RT. The eluate was filtered through pierce spin columns (Thermo, #69725) to remove remaining ConA beads, and then, the filtrate was transferred to a 1.5-ml protein LoBind tube. The elution step was repeated with another 500 μl elution buffer and combined with the first eluate. 50 μl of magnetic DBCO beads (Jena bioscience, Cat #CLK-1037-1) was washed twice with mass spec grade water and added to the eluate. Sodium deoxycholate (SDC) was added to a final concentration of 0.1% (w/v) to prevent clumping/sticking of the beads (except otherwise noted). The click reaction was performed while shaking overnight at 4°C on an Eppendorf Thermo-Mixer R shaker to covalently couple metabolically labeled glycoproteins to the magnetic beads. The next day, beads were washed three times with 1 ml 1% SDS buffer (100 mM Tris–HCl pH 8.5, 1% SDS, 250 mM NaCl), three times with 1 ml 8 M UREA buffer (8 M Urea in 100 mM Tris–HCl pH 8.5), and three times with 1 ml 20% (v/v) acetonitrile. Beads were retained with a magnetic rack (DynaMag-2, Thermo Scientific). After each step, samples were resuspended

briefly by shaking at room temperature. Beads were transferred to a new 1.5-ml low binding tube using $2 \times 500$ μl mass spec grade water. Beads were retained in a magnetic rack, and the supernatant was removed. Protein disulfide bonds were reduced in 50 μl of 10 mM dithiothreitol (DTT) in 100 mM ammonium bicarbonate (ABC) for 30 min at 37°C. Afterward, the supernatant was discarded. Alkylation of cysteines was performed using 50 μl of 55 mM iodoacetamide (IAA) in 100 mM ABC for 30 min and 20°C in the dark. The supernatant was discarded, and beads were washed twice with 100 μl of 100 mM ABC. The protein digestion was performed by adding 0.2 μg LysC (Promega) in 50 μl of 100 mM ABC for 3 h at 37°C followed by overnight trypsin digestion using 0.2 μg of trypsin (Promega) per sample in 100 mM ABC without 0.1% SDC. The supernatant containing the tryptic peptides was transferred to a 0.5-ml protein LoBind tube. Beads were washed twice with 100 mM ABC without 0.1% SDC and added to the same tube. Each sample was acidified with 50 μl of 8% FA and incubated for 20 min at 4°C. Precipitated SDC was removed by centrifugation at 18,000 *g* for 20 min at 4°C. Peptides were cleaned up using C18 stage tips as previously described (Rappsilber *et al*, 2003). Dried peptides were resuspended in 18 μl 0.1% formic acid (FA), and 2 μl of 1:10 diluted iRT peptides (Biognosys, Ki-3002-1) was spiked into the samples.

### Mass spectrometry

The LC-MS/MS analyses were performed on an EASY-nLC 1200 UHPLC system (Thermo Fisher Scientific) which was online coupled with a NanoFlex ion source equipped with a column oven (Sonation) to a Q Exactive™ HF Hybrid Quadrupole-Orbitrap™ mass spectrometer (Thermo Fisher Scientific). 8 μl per sample was injected. Peptides were separated on a 30-cm self-made C18 column (75 μm ID) packed with ReproSil-Pur 120 C18-AQ resin (1.9 μm, Dr. Maisch GmbH). For peptide separation, a binary gradient of water and 80% acetonitrile (B) was applied for 120 min at a flow rate of 250 nl/min and a column temperature of 50°C: 3% B 0 min; 6% B 2 min; 30% B 92 min; 44% B 112 min; and 75% B 121 min.

Data-dependent acquisition (DDA) was used with a full scan at 120,000 resolution and a scan range of 300–1,400 *m/z*, automatic gain control (AGC) of $3 \times 10^6$ ions, and a maximum injection time (IT) of 50 ms. The top 15 most intense peptide ions were chosen for collision-induced dissociation (CID) fragmentation. An isolation window of 1.6 *m/z*, a maximum IT of 100 ms, and AGC of $1\times10^5$ were applied, and scans were performed with a resolution of 15,000. A dynamic exclusion of 120 s was used.

Data-independent acquisition (DIA) was performed using a MS1 full scan followed by 20 sequential DIA windows with variable width for peptide fragment ion spectra with an overlap of 1 *m/z* covering a scan range of 300–1,400 *m/z*. Full scans were acquired with 120,000 resolution, AGC of $5 \times 10^6$, and a maximum IT time of 120 ms. Afterward, 20 DIA windows were scanned with a resolution of 30,000 and an AGC of $3 \times 10^6$. The maximum IT for fragment ion spectra was set to auto to achieve optimal cycle times. The *m/z* windows were chosen based on the peptide density map of the DDA run of a representative hiSPECS sample and optimized in a way that allowed the detection of 8 points per peak. The following window widths were chosen according to the peptide density map (Fig 1B):

| Window # | 1 | 2 | 3 | 4 | 5 | 6 | 7 | 8 | 9 | 10 | 11 | 12 | 13 | 14 | 15 | 16 | 17 | 18 | 19 | 20 |
|---|---|---|---|---|---|---|---|---|---|---|---|---|---|---|---|---|---|---|---|---|
| Window width (*m/z*) | 85 | 40 | 30 | 28 | 26 | 25 | 24 | 24 | 24 | 24 | 25 | 27 | 28 | 29 | 34 | 38 | 46 | 61 | 151 | 352 |

The following libraries were used for the different DIA experiments of this manuscript using the search engine platform MaxQuant and Spectronaut Pulsar X version 12.0.20491.14:

| Library | Unique modified peptides | Unique peptide precursors | Protein groups | Used in: |
|---|---|---|---|---|
| hiSPECS neuron | 4,585 | 5,957 | 646 | Tables EV1, EV6 |
| hiSPECS brain cell types | 12,695 | 15,886 | 1,540 | Table EV2 |
| Murine CSF | 20,100 | 24,985 | 2,550 | Table EV7 |
| hiSPECS slices | 2,675 | 3,375 | 431 | Table EV8 |

### BACE1 inhibitor treatment

The BACE inhibitor C3 (β-secretase inhibitor IV, Calbiochem, Cat #565788, Sigma-Aldrich) or DMSO vehicle control was added in parallel to the ManNAz, at a final concentration of 2 μM, to the neurons for 48h (Kuhn *et al*, 2012).

### LPS treatment of brain slices

Organotypic brain slices were cultured for 2 weeks after the isolation process, followed by a 48-h ManNAz labeling step. Next, the brain slices were treated for 6 h with LPS (500 ng/ml) followed by a 24-h collection window of the conditioned media. Serum supplements were not added during the LPS treatment and the collection period to maximize the inflammatory response. However, additional ManNAz was added during all steps.

### Antibodies

The following antibodies were used for Western blotting: mouse monoclonal anti-ADAM22 (UC Davis/NIH NeuroMab Facility Cat #75-093), rat monoclonal anti-SEZ6 (Pigoni *et al*, 2016), goat polyclonal anti-CD200 (R and D Systems Cat #AF2724), and rabbit polyclonal anti-calnexin (Enzo Life Sciences Cat #ADI-SPA-860).

### ELISA

CD200 ectodomain levels in the supernatant of primary neurons were measured and quantified using the following ELISA Kit according to the supplier's manual: Mouse CD200 ELISA Kit (LSBio Cat #LS-F2868). A volume of 250 μl from the total of 1 ml undiluted media of 1.5 million primary cortical neurons cultured for 48h was used per technical replicate. The standard curve provided with the kit was used, and neuronal media, which was not cultured with cells, were used as a blank value.

### Western blot analysis

Cells were lysed in STET buffer (50 mM Tris, pH 7.5, 150 mM NaCl, 2 mM EDTA, 1% Triton X-100) and incubated for 20 min on ice with intermediate vortexing. Cell debris as well as undissolved material was removed by centrifugation at 20,000 *g* for 10 min at 4°C. The protein concentration was determined using the BC assay kit of Interchim (UP40840A) according to the manufacturer's instructions. Samples were boiled for 10 min at 95°C in Laemmli buffer and separated on self-cast 8%, 10%, or 12% SDS–polyacrylamide gels. Afterward, proteins were transferred onto nitrocellulose membranes using a Bio-Rad Wet/Tank Blotting system. The membranes were blocked for 20 min in 5% milk in 1xPBS with 1% Tween, incubated overnight at 4°C with the primary antibody, 1 h at room temperature with the secondary antibody, and developed using an ECL prime solution (GE Healthcare, RPN2232V1).

### Cerebrospinal fluid (CSF) sample preparation

In-solution digestion of 5 μl CSF samples was performed as previously described (Pigoni *et al*, 2016). Dried peptides were dissolved in 18 μl 0.1% FA and 2 μl 1:10 diluted iRT peptides.

### Quantification and statistical analysis

In general, statistical details can be found in the figure legends including statistical tests and N number used.

### Raw data analysis of mass spectrometry measurements

Data-dependent acquisition raw data were analyzed with MaxQuant (version 1.5.5.1 or 1.6.1.0) using the murine UniProt reference database (canonical, downloaded on 17.01.2018, which consisted of 16,954 proteins) and the Biognosys iRT peptide database for label-free quantification (LFQ) and indexed retention time spectral library generation. Default settings were chosen; however, the minimal peptide length was set to six. Two missed cleavages were allowed, carbamidomethylation was defined as a fixed modification, and N-termini acetylation as well as oxidation of methionines was set as variable modifications. The false discovery rate (FDR) was set to less than 1% for protein and peptide identifications. The results of the MaxQuant analysis were used to generate DIA spectral libraries of proteins in Spectronaut Pulsar X (Biognosys). Data generated with DIA were analyzed using Spectronaut Pulsar X (Biognosys) using the self-generated spectral libraries applying default settings: quantification on the MS2 level of the top N (1–3) peptide spectra and a FDR of 1%.

### Bioinformatic analysis: Data Pre-processing and Normalization

For the hiSPECS brain secretome resource, the raw dataset was filtered to retain only the proteins that were consistently quantified in at least 5 of the 6 biological replicates (5/6 or 6/6) in any of the four

cell types. This yielded a total of 995 out of 1,083 quantified proteins. The data were further processed using Perseus (Version 1.6.6.0). The LFQ values were $\log_2$-transformed, and the Pearson correlation coefficients between all samples were determined. An imputation procedure was employed by which missing values are replaced by random values of a left-shifted Gaussian distribution (shift of 1.8 units of the standard deviation and a width of 0.3).

### PCA and UMAP

Principal component analysis and uniform manifold approximation and projection (UMAP) (Diaz-Papkovich *et al*, 2019) were performed to visualize the relationships between the cell types and between the biological replicates. UMAP is a fast non-linear dimensionality reduction technique that yields meaningful organization and projection of data points. UMAP has the advantage of assembling similar individuals (or data points) while preserving long-range topological connections to individuals with distant relations.

### Differential abundance analysis

In order to determine differentially abundant proteins using pairwise comparisons of all four brain cell types, we employed protein-wise linear models combined with empirical Bayes statistics [implemented in the R package Limma (Ritchie *et al*, 2015), similarly to (Kammers *et al*, 2015)]. A protein was considered as differentially abundant (DA) in the different brain cell types if the Bonferroni-corrected *P*-value was < 0.05 and the $\log_2$ fold change ≥ 2. The $\log_2$ fold change of 2 was set to reduce the number of false positives due to data imputation.

### Pathway enrichment score

Proteins were considered to be mainly secreted from one cell type if proteins were identified in at least 5 out of 6 biological replicates and fulfilled one of following criteria: (i) Proteins were only detected in two or less biological replicates in another cell type or (ii) proteins were at least fivefold enriched in a pairwise comparison to all three other cell types. Functional annotation clustering of proteins specifically secreted from each cell type was performed with DAVID 6.8 (da Huang *et al*, 2009a,b) using all 995 robustly quantified proteins in the hiSPECS secretome resource as the background. The top three gene ontology terms for biological process (GOTERM_BP_FAT) were picked for visualization based on the enrichment score using medium classification stringency. The same settings were chosen to identify functional annotation clusters of all proteins in the hiSPECS library identified compared with the whole mouse proteome for the gene ontology term cellular compartment (GOTERM_CC_FAT).

### Comparison of secretome proteins to their relative abundances in the corresponding cell lysates using a published database

We compared our cell type-specific secretome proteins to the corresponding cell type-specifically expressed proteins identified in cell lysates of the same four cultured primary brain cell types in a previous proteomic study (Sharma *et al*, 2015). The data from their study containing the LFQ values of the individual biological replicates of the brain cell lysate were downloaded (Sharma *et al*: Appendix Table S1) and processed using Perseus. Proteins detected in at least one cell type with two biological replicates were considered, and missing values were imputed as described before

(replacement of missing values from left-shifted normal distribution, 1.8 units of the standard deviation, and a width of 0.3). Proteins with 2.5-fold higher protein levels in one cell type as compared to all other three cell types were considered specific. On the basis of this analysis, we defined two categories: (i) proteins that were specifically enriched in one cell type both in cell lysates and the corresponding cell secretome and (ii) cell type-specifically secreted proteins that were not enriched in the corresponding lysate. The second category indicates a possible secretome-specific mechanism, such as the selective secretion/shedding by a protease in the given cell type.

### Interactions with cell lysate proteins detected by Sharma et al

To find the relationships between the proteins in the secretome and those from the cell lysate, we searched for interacting partners of the proteins specifically enriched in the secretome of a specific cell type and the proteins from the cell lysate as determined by Sharma *et al* (2015). We downloaded the mouse protein–protein interaction network (PPIN) from BioGRID (Chatr-Aryamontri *et al*, 2015) and additional binary interaction data from UniProt (The UniProt Consortium, 2018).

### Disease association

To determine whether the proteins significantly secreted from one cell type with respect to the other 3 cell types are linked to neurodegenerative diseases, we searched for curated gene–disease associations (GDA) from DisGeNET (Pinero *et al*, 2017). Our search list contained 31 known diseases of the nervous system. We set the evidence index (EI) to 0.95. An EI of 1 indicates that all the available scientific literature supports the specific GDA.

# Data availability

The proteomic resource is available for the public on the ProteomeXchange Consortium via the PRIDE Archive (project accessions: PXD018171; http://www.ebi.ac.uk/pride/archive/projects/PXD018171 and PXD020503; http://www.ebi.ac.uk/pride/archive/projects/PXD020503).

**Expanded View** for this article is available online.

### Acknowledgements

We thank Felix Meissner and Jürgen Cox for helpful comments on this manuscript, and Anna Daria for support with the *ex vivo* model. This work was supported by the Deutsche Forschungsgemeinschaft (German Research Foundation) within the framework of the Munich Cluster for Systems Neurology (EXC 2145 SyNergy, project ID 390857198) and the BMBF through project CLINSPECT-M and JPND PMG-AD. JT was supported by a Boehringer Ingelheim Fonds (BIF) PhD fellowship. Funds have been provided to ST by the Alzheimer Forschung Initiative e.V. (project number 18014). Open access funding enabled and organized by Projekt DEAL.

### Author contributions

JT, SAM, and SFL conceptualized, validated, and wrote the manuscript; JT and SAM performed the experiments, involved in methodology, investigation, and formal analysis; JT, SAM ESK, JZ, DF, and GG involved in formal analysis; LSM, ST, MSu, MSi, and GJ provided resources; JT visualized the study; SAM and SFL supervised the study; and SFL acquired funding.

## Conflict of interest

The authors declare that they have no conflict of interest.

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
