## [Review Process File · The EMBO Journal]

Quantitative proteomics establishes the cell type-resolved mouse brain secretome

Johanna Tüshaus, Stephan Müller, Evans Kataka, Jan Zaucha, Laura Sebastian Monasor, Minhui Su, Gökhan Güner, Georg Jocher, Sabina Tahirovic, Dmitrij Frishman, Mikael Simons, and Stefan Lichtenthaler

DOI: [10.15252/embj.2020105693](https://doi.org/10.15252/embj.2020105693)

Corresponding author: Stefan Lichtenthaler (stefan.lichtenthaler@dzne.de)

Review Timeline:

Submission Date:	20th May 20
Editorial Decision:	2nd Jul 20
Revision Received:	24th Jul 20
Editorial Decision:	7th Aug 20
Revision Received:	12th Aug 20
Accepted:	14th Aug 20

Editor: Karin Dumstrei

Transaction Report:

Dear Stefan,

Thank you for submitting your manuscript to The EMBO Journal. Your study has now been seen three referees and their comments are provided below.

As you can see below, the referees find the reported method and the brain secretome analysis a very valuable resource for the field. Given these comments, I would therefore like to invite you to submit a suitably revised manuscript that takes the referees comment into consideration. The raised issues should be fairly straightforward to address and I don't anticipate that the revisions will involve major efforts. Let me know if there is anything specific to discuss - happy to do so. Regarding referee #3's comments: we are considering the submission as a resource and don't need further experiments to explore mechanism or biology (ref #3 point 3).

I thank you for the opportunity to consider your work for publication. I look forward to your revision.

with best wishes

Karin

Karin Dumstrei, PhD
Senior Editor
The EMBO Journal

When assembling figures, please refer to our figure preparation guideline in order to ensure proper formatting and readability in print as well as on screen:
<http://bit.ly/EMBOPressFigurePreparationGuideline>

Further information is available in our Guide For Authors:

The revision must be submitted online within 90 days; please click on the link below to submit the revision online before 30th Sep 2020.

Link Not Available

Referee #1:

The collection of proteins secreted from cells is referred to as the secretome, and is implicated in neurodegenerative diseases, and include APOE and TREM2. This study is relevant, as the source of secreted proteins in cerebrospinal fluid can't be traced to specific cell types. Current limitations of scale and specificity limit secretome analysis of brain cell fractions. Tüshaus et al develop a new technique, 'high performance secretome protein enrichment with click sugars', aka hiSPECS, to allow for primary cell analysis in single animal sections. MS analysis indicated a 99.9% collection of ectodomain regions for collected tryptic proteins that required only 1/40th of the sample size and 1/5th of the preparation time of standard protocols. The authors used hiSPECS to determine the cell-specific secretomes of mouse brain cell cultures, slices, and CSF.

Using hiSPECS, novel interactions between cell types were also observed, for example oligodendrocyte secretion of adiponectin and reception by cadherin13 in neurons. The authors also demonstrated that different cell type secretomes have different compositions of ectodomain shedding, with neurons shedding the largest fraction. These cleaved fractions include Sez6 and other synaptically relevant signaling proteins, indicating the functional relevance to synaptic transmission. Conversely, in astrocytic and microglial secretomes, the majority was comprised of secreted soluble proteins, consistent with established cellular functions. BACE1, a primarily neuronal

protein, generated cleavage products that were selectively enriched in the neuronal secretome. Primary neurons were treated with a BACE inhibitor (C3) and 29 membrane protein levels were reduced. CSF from wildtype mice was able to identify a large fraction of protein groups, and ~65% were mapped to the corresponding cell type from primary culture. Surprisingly, neurons accounted for the majority of the CSF secretome. Lastly, organotypic cotric-hippocampal brain sections were examined with hiSPECS and mapped onto the primary culture results, as well as treated with LPS to induce inflammation. LPS treatment showed all four cell types contribute to the inflammatory response.

The work presented in this manuscript is a useful tool in starting to determine the cell-specific types, locations, and functionalities of the secretome. However, in its present form, this manuscript raises a number of comments, as outlined below, which should be addressed:

- The manuscript appears to use iSPECS and hiSPECS interchangeably. Please clarify if these are two different techniques, and if so how, or refer to the technique with the same name throughout the manuscript.
- A more thorough explanation of DDA vs DIA and the implications to hiSPECS would be appreciated.
- Figure 2D appears to show a correlation with astrocytes and oligodendrocytes. A brief commentary on this would be appropriate.
- NCAM2 is shown to be selectively enriched in oligodendrocytes in S2, however it is not reported as one of the top 50 cell specifically-enriched proteins in S4A. Please explain.
- Given that roughly half of secreted proteins are cell-type specific, and the conclusion that this accounts for functional differences (lines 202-207), how would a primary culture of one cell type be influenced by another? Ex. Neurons treated with oligodendrocyte secretome.
- C3 is also an inhibitor of BACE2, which should be explicitly addressed in the manuscript. Indeed, Pignoni et al 2016 showed BACE2 deficient mice also failed to cleave CD200.
- How does C3-mediated BACE alterations of the secretome shift upon neuronal induction of potentiation (ex. chemically by forskolin)?
- A profile of hippocampal neurons versus cortical neurons would be very informative, especially for future work in AD model mice or treatments with C3.
- An important caveat to this study is the different behavior and expression profiles of primary culture versus mixed cultures versus tissues. It is not certain that primary astrocyte cultured cells would express the same secretome as astrocytes in co-culture or in tissue. Therefore, these results should be taken and discussed with that caveat in mind.

Other minor concerns:

- The title is somewhat redundant. Tüshaus et al should rephrase.
- Figure 5C Western blot contains bubbles in the bands of interest

Referee #2:

General overview:

Tüshaus et al. describe a second-generation proteomic method that enables the characterization of secreted proteins, i.e., the secretome. Based on the original method (SPECS, secretome protein enrichment with click sugars; Kuhn et al, 2012), the improved technology-now called hiSPECS-is based on several technological improvements (e.g., the use of concavalin to enrich for glycoproteins away from serum-proteins, on-bead copper-free click chemistry and trypsin digest, and MS detection using DDA and DIA modes). With these improvements, the authors demonstrate that the culture volumes can be significantly smaller for a hiSPECS experiment (1 million primary cells vs. 40 million for SPECS) and then characterize the secretomes of four different cell types (astrocytes, microglia, neurons, oligodendrocytes). Combined with other published proteomic analyses (e.g., Sharma et al 2015), the authors use their data to draw conclusions about specific protein-protein interaction partners between different cell types, how specific cell types contribute to the secretome within CSF (which interestingly does not correlate with protein expression), and lastly, how LPS stimulates different protein responses in different cell types under neuroinflammatory conditions. This is a nicely constructed study that adds value to the proteomic characterization of brain cell types and provides molecular clues for how different cell types contribute to the complex neuronal environment. In comparison to its parent method SPECS, hiSPECS offers a more accessible proteomic platform that can be readily applied to a variety of different cell types under many biological contexts.

Major comments:

-In the volcano plot comparing C3 vs. control treatment of neurons, the authors identify several downregulated shedded proteins. Some of these shedded proteins (e.g., VCAM1, also observed in astrocytes) are also present in the secretive of other cell types reported in Supplementary Table 2. If these cells lack BACE1, can the authors comment on what other proteases could be responsible for these shedding events?

-Using the BACE1 inhibitor C3, the authors find that 29 neuronal proteins undergo less shedding. The dataset includes ADAM22 which can act as a soluble protease after cleavage from its pro-domain. Can the authors distinguish BACE1 substrates from ADAM22 substrates?

-In Figure 5c, there are arrows pointing to two bands for flADAM22. The datasheet for the corresponding anti-ADAM22 antibody used does show a predominant single band and then a fainter lower band beneath. Full length CD200 also appears to run as two bands. Can the authors explain why there are two bands for flADAM22/flCD200 and whether or not reducing agent was added to these samples? There are also arrows pointing to the flADAM22 bands in Fig. 5c and there should be a corresponding description in the figure caption or main text.

Minor comments:

-iBaQ is used in Figure 1D and should be defined in the main text.

-Can the authors comment on the glycosylation differences between the different cell types? Specifically, do the authors know if there are more or less sialic acid sugars that may allow different amounts of azido-labeling and how this may impact their datasets.

-It is interesting that there is a significantly higher proportion of membrane proteins in the list of unique proteins specific to neuronal cells and that this is not the case for the other three cell types. The authors suggest that this is due to different shedding activity levels. As noted by Sharma et al 2015, these four cell types all have a unique, enriched pattern of integral membrane proteins. The authors also comment that integral membrane proteins are not part of their shedded dataset, consistent with the observation that they did not see the same enrichment. Are there other proteomic studies that have characterized the membrane proteomes for these cell lines? If so, are there differences in the abundance and diversity of membrane proteins that the authors are aware of among these cells.

Referee #3:

This paper describes an important technical advance that enables analysis of secreted glycoproteins in cultured cells by proteomics. I think this work is exciting and topical, but feel that there are key questions that I as a referee with a non-proteomics background can only partly comment on.

1. Is the development of hiSPECS a highly significant advance compared to the previously described SPECS method, such that it qualifies for publication in EMBO J.? I cannot assess this question but a proteomics expert might be able to comment.
2. Is the present paper as a pure technical report appropriate for EMBO J.? This is more an editorial than a scientific question, but I note that the previous study on SPECS by the same authors, published in EMBO J., identified BACE substrates as a biological application whereas the current paper lacks a major biological theme.
3. What interesting new findings does the current study report? The present paper demonstrates that neurons but not other cell types appear to produce a secretome that includes a large number of ECDs from cleaved cell surface proteins. This is interesting, as it suggests that surface proteases are much more active in neurons than other cell types. If the authors could dig a bit deeper on this theme and obtain independent evidence for this observation as well as explore its mechanism and physiological significance, this paper would be much more appropriate for EMBO J. than in its current version.

My overall impression is that this interesting paper would benefit from a deeper exploration of the biology. I hope my comments will be helpful.

We thank the Referees for their encouraging and constructive reviews. A point-by-point response to each of their comments is provided below.

Reviewer #1

The collection of proteins secreted from cells is referred to as the secretome, and is implicated in neurodegenerative diseases, and include APOE and TREM2. This study is relevant, as the source of secreted proteins in cerebrospinal fluid can't be traced to specific cell types. Current limitations of scale and specificity limit secretome analysis of brain cell fractions. Tüshaus et al develop a new technique, 'high performance secretome protein enrichment with click sugars', aka hiSPECS, to allow for primary cell analysis in single animal sections. MS analysis indicated a 99.9% collection of ectodomain regions for collected tryptic proteins that required only 1/40th of the sample size and 1/5th of the preparation time of standard protocols. The authors used hiSPECS to determine the cell-specific secretomes of mouse brain cell cultures, slices, and CSF.

The work presented in this manuscript is a useful tool in starting to determine the cell-specific types, locations, and functionalities of the secretome. However, in its present form, this manuscript raises a number of comments, as outlined below, which should be addressed:

1. **The manuscript appears to use iSPECS and hiSPECS interchangeably. Please clarify if these are two different techniques, and if so how, or refer to the technique with the same name throughout the manuscript.**

We are sorry for the typos that occurred. It is always the same technique. We have corrected the typos and now use hiSPECS throughout the manuscript.

2. **A more thorough explanation of DDA vs DIA and the implications to hiSPECS would be appreciated.**

Data dependent acquisition (DDA) and data independent acquisition (DIA) are two different modes in which proteomic data can be acquired. DDA, the classical way of performing proteomic discovery experiments, relies on a first mass spec scan (MS1 scan) where the most abundant (tryptic) peptides derived from the proteins in the sample are selected and then fragmented for sequencing and are then being analyzed in a second mass spec scan (MS2 scan). In contrast, DIA is not limited to the most abundant peptides for the sequencing MS2 scans, but fragments and detects all peptides within a defined m/z window. In this way DIA is highly advantageous for the detection of low abundant peptides, which are typical for secretome samples. However, DIA is more complicated to set up. Usually, it requires a spectral library based on DDA runs containing information of indexed retention times and the fragment ion patterns of peptides due to the complexity of the MS2 spectra, which are recorded for broad m/z windows (Ludwig et al., 2018). In the hiSPECS protocol DIA is the method of choice because it allows deeper proteome coverage and therefore a more complete hiSPECS secretome resource (compare Figure 1D).

We included the following two paragraphs into the manuscript (See page 6 line 98-103 & page 7 line 112-116):

“DDA and DIA are two different modes in which proteomic data can be acquired. In contrast to DDA, DIA is not limited to the most abundant peptides for the subsequent peptide sequencing and identification, but fragments and detects all peptides within a defined m/z window, which

may be particularly advantageous for the detection of lower abundant peptides (Bruderer et al., 2015, Gillet et al., 2012, Ludwig et al., 2018, Sebastian Monasor et al., 2020).”

“Overall, DIA extended the dynamic range for secretome protein quantification by almost one order of magnitude, which was evaluated based on intensity based absolute quantification (iBAQ) values representing a rough estimate of molar protein abundances within a sample (Schwanhäusser et al., 2011)(Fig 1D). Due to the superiority of DIA over DDA in regards of proteome coverage and reproducibility for secretome analysis, we focused on hiSPECS DIA throughout the manuscript.”

3. Figure 2D appears to show a correlation with astrocytes and oligodendrocytes. A brief commentary on this would be appropriate.

Figure 2D shows the Pearson correlation coefficients of the log₂ transformed LFQ values (corresponding to the intensity of the proteins) of the six biological replicates of all four cell types included in the iSPECS secretome resource. The biological replicates within one cell type reveal very high correlations. In contrast, the correlations between different cell types is lower. The slightly higher degree of correlation for astrocytes with oligodendrocytes versus neurons may go back to their common lineage (Hirano & Goldman, 1988). In line with our findings on the secretome, the lysate proteomes of oligodendrocytes and astrocytes were also reported to show a higher correlation compared to neurons (Sharma et al., 2015).

We included the following paragraph into the manuscript on page 9 line 158-163:

“Pairwise correlation revealed that some secretomes correlate more closely than others (Fig. 2D). For example, the secretome of oligodendrocytes correlated to a higher degree with the secretome of astrocytes than with the secretome of neurons, which was also observed for the cell lysate proteome of the same brain cell types (Sharma et al., 2015) and may reflect their common origin from the glia lineage (Hirano & Goldman, 1988).”

4. NCAM2 is shown to be selectively enriched in oligodendrocytes in S2, however it is not reported as one of the top 50 cell specifically-enriched proteins in S4A. Please explain.

Of note, S2 is now Figure EV2 and S4A is now Figure EV3A.

The hiSPECS secretome resource revealed NCAM2 to be selectively enriched in the supernatant of oligodendrocytes compared to the other 3 brain cell types (Figure EV2A, Table EV4). In Figure EV2A we picked 3 protein examples for each cell type (astrocytes, microglia, neurons, oligodendrocytes) which show enrichment in the lysate (Sharma et al. data) AND the secretome as part of the quality control of our cell cultures used for setting up the resource. Thus, the increased secretion of NCAM2 from oligodendrocytes may be due to its increased expression in this cell type compared to the other ones. In contrast, the aim of Figure EV3 is different. It visualizes the top 50 changed proteins considering the fold-changes in a pairwise comparison of all 995 proteins included in the iSPECS resource library after data imputation. This may also include proteins that show cell type-specific secretion, but are similarly expressed among different cell types. NCAM2 is not among the top 50 changed proteins even though it is significantly selectively secreted from oligodendrocytes. This means that at least 50 other proteins of the iSPECS resource revealed even more pronounced fold changes for a specific cell type compared to NCAM2.

In the manuscript, we made this point clearer by adding a sentence to the legend of Fig EV2C:

“Known cell type-specific marker proteins are highlighted for each cell type which reveal a strong enrichment in the lysate and secretome of the primary brain cells verifying the quality and comparability of the primary cultures. For example, the ectodomain of the membrane protein NCAM2 (with an y-axis value of 3 in the log2 scale) is secreted about 8-fold more from oligodendrocytes compared to the median of the other three cell types and also expressed at a higher level in this cell type compared to the median of the other three cell types.”

5. **Given that roughly half of secreted proteins are cell-type specific, and the conclusion that this accounts for functional differences (lines 202-207), how would a primary culture of one cell type be influenced by another? Ex. Neurons treated with oligodendrocyte secretome.**

This is an interesting experimental suggestion. Because our manuscript is considered as a resource, this experiment is not required at this point, in agreement with the decision letter of the editor. However, following the suggestion of this reviewer, we have a sentence in the discussion (line 423) stating that the secretome may change upon cocultures: “Additionally, the secretome may change upon cell stimulation, such as during neuronal activity or inflammation or when different cell types are cocultured or taken into three-dimensional culture systems”.

6. **C3 is also an inhibitor of BACE2, which should be explicitly addressed in the manuscript. Indeed, Pignoni et al 2016 showed BACE2 deficient mice also failed to cleave CD200.**

As suggested we added the following sentence to the manuscript (page 12, line 240-242):

“C3 can also inhibit BACE2, a close homolog of BACE1, but in contrast to BACE1, BACE2 is very little expressed in neurons (Voytyuk et al., 2018).”

Regarding your other point “Pignoni et al 2016 showed BACE2 deficient mice also failed to cleave CD200”: That study did not show that CD200 cleavage is abolished in BACE2 single KO mice. Instead, they performed whole proteome analysis of CSF samples from BACE1 and BACE2 double knockout (DKO), but not BACE1 or BACE2 single knockout mice compared to wildtype mice. The tryptic peptide of CD200 was exclusively found in wildtype CSF, but not in BACE1+2 DKO CSF. We agree that this does not rule out the possibility that CD200 may also be cleaved by BACE2. However, in our secretome resource, CD200 was found to be exclusively secreted from neurons and it is well known that BACE1 but not BACE2 is highly expressed in neurons (Voytyuk et al., 2018). Together these results indicate that it is very likely that the tryptic peptide found in wildtype CSF is a product of proteolytic cleavage of CD200 by BACE1 and not BACE2 in neurons. Of note, the CSF of single BACE2 knock-out mice was only used for immunoblot validation of the substrate SEZ6, but not of CD200, in Figure 7D of the Pignoni et al. publication.

7. **How does C3-mediated BACE alterations of the secretome shift upon neuronal induction of potentiation (ex. chemically by forskolin)?**

This is an interesting experimental suggestion that may be carried out in a future study. Because our manuscript is considered as a resource, this experiment is not required at this point, in agreement with the decision letter of the editor.

8. **A profile of hippocampal neurons versus cortical neurons would be very informative, especially for future work in AD model mice or treatments with C3.**

As suggested, we prepared hippocampal neurons, determined their secretome and compared it the one of cortical neurons. We observed a very high degree of correlation ($R^2=0.85$) between

both secretomes (new Fig EV5D) and also demonstrate in a Venn diagram that the vast majority of proteins (356 out of 434) were detected in both secretomes (new Fig EV5E). We did an additional experiment, where we prepared hippocampal neurons and treated them with the BACE inhibitor C3 or DMSO as a control and then determined the C3-dependent changes in the secretome. The results are displayed in a volcano plot (new Fig EV5F) and demonstrate that C3 induces similar changes in the secretome as observed for the cortical neuronal secretome in Fig 5. We conclude that BACE1 substrates are very similar in cortical and hippocampal neurons.

The new experiments are now described in the results' section on page 12-13.

- 9. An important caveat to this study is the different behavior and expression profiles of primary culture versus mixed cultures versus tissues. It is not certain that primary astrocyte cultured cells would express the same secretome as astrocytes in co-culture or in tissue. Therefore, these results should be taken and discussed with that caveat in mind.**

This is an important point that we address in the discussion. The paragraph (page 19) starts with: "The secretome of the four major brain cell types and the *ex vivo* tissue identified here represents a snapshot of the total brain secretome and more secreted proteins are known or likely to exist." Then continues a bit later with: "For example, it is known that microglia can partially change their expression profile when taken into the culture, which may consequently affect the secretome (Gosselin et al., 2017). Additionally, the secretome may change upon cell stimulation, such as during neuronal activity or inflammation or when different cell types are cocultured or taken into three-dimensional culture systems (Stiess et al., 2015)."

We did not specifically mention astrocytes here, but do cite a paper (Stiess et al. 2015), which studied the astrocyte secretome in a coculture approach.

- 10. Other minor concerns: The title is somewhat redundant. Tüshaus et al should rephrase.**

We agree: we had twice the term "secretome" in the title replaced the first occurrence with "proteomics". The new title is now more informative and reads as: "Quantitative proteomics establishes the cell type-resolved mouse brain secretome".

- 11. Other minor concerns: Figure 5C Western blot contains bubbles in the bands of interest**

We agree that it would be nicer to have blots without bubbles. Unfortunately, blots are not always perfect. But, importantly, the bubbles are only seen on the lysate and not the secretome blots and only in 1 out of 3 replicates. Even more importantly, we had 6 biological replicates for our quantification and we add for the reviewer the other 3 replicates not shown in figure 5, which gave the same result. Together, we feel that an occasional bubble does not interfere the quantification, because the variation between experiments is larger than an effect of a bubble (see quantification in Fig. 5D).

Figure 1: Replicate 4-6 of Western blot analysis shown in Fig 5C of neuronal lysate either treated or untreated (DMSO) with the B1-inhibitor C3. Western blot analysis of ADAM22, CD200 and calnexin are shown.

Reviewer #2

General overview:

Tüshaus et al. describe a second-generation proteomic method that enables the characterization of secreted proteins, i.e., the secretome. Based on the original method (SPECS, secretome protein enrichment with click sugars; Kuhn et al, 2012), the improved technology-now called hiSPECS-is based on several technological improvements (e.g., the use of concavalin to enrich for glycoproteins away from serum-proteins, on-bead copper-free click chemistry and trypsin digest, and MS detection using DDA and DIA modes). With these improvements, the authors demonstrate that the culture volumes can be significantly smaller for a hiSPECS experiment (1 million primary cells vs. 40 million for SPECS) and then characterize the secretomes of four different cell types (astrocytes, microglia, neurons, oligodendrocytes). Combined with other published proteomic analyses (e.g., Sharma et al 2015), the authors use their data to draw conclusions about specific protein-protein interaction partners between different cell types, how specific cell types contribute to the secretome within CSF (which interestingly does not correlate with protein expression), and lastly, how LPS stimulates different protein responses in different cell types under neuroinflammatory conditions. This is a nicely constructed study that adds value to the proteomic characterization of brain cell types and provides molecular clues for how different cell types contribute to the complex neuronal environment. In comparison to its parent method SPECS, hiSPECS offers a more accessible proteomic platform that can be readily applied to a variety of different cell types under many biological contexts.

Major comments:

- 2.1. **In the volcano plot comparing C3 vs. control treatment of neurons, the authors identify several downregulated shedded proteins. Some of these shedded proteins (e.g., VCAM1, also observed in astrocytes) are also present in the secretive of other cell types reported in Supplementary Table 2. If these cells lack BACE1, can the authors comment on what other proteases could be responsible for these shedding events?**

In general the ectodomain shedding of membrane proteins is mediated by more than 30 different proteases. Importantly, one single protein may be shed by distinct proteases (e.g. APP by ADAM10 and BACE1 and delta- and eta-secretase) and one single protein may be preferentially shed by one protease in one cell type but by another protease in another cell type. Again, APP (but also CHL1) serves as an excellent example. Shedding of APP in neurons is predominantly mediated by BACE1 and only to a low extent by ADAM10, whereas in non-neuronal cells the relative extents are the opposite. The underlying reason is presumably the expression level of the given protease. BACE1 is highly expressed in neurons, but only at low levels in other cell types. For VCAM1 the situation appears similar. With C3 VCAM1 shedding is inhibited by about 40% which demonstrates that VCAM1 in neurons is clearly also shed by other proteases. And these other proteases may be the relevant ones in the other brain cell types, where little BACE1 is expressed. For example, the protease ADAM17 has been suggested to cleave VCAM1 (Garton, Gough et al., 2003). And we have previously shown that VCAM1 can also be shed by BACE2, a BACE1 homolog, that is not expressed in neurons, but can be expressed in other cell types of the brain (Voytyuk, Mueller et al., 2018). Thus, it is very likely that VCAM1, secreted from other brain cell types, is shed by ADAM17 and BACE2 or potentially even additional proteases.

- 2.2. **Using the BACE1 inhibitor C3, the authors find that 29 neuronal proteins undergo less shedding. The dataset includes ADAM22 which can act as a soluble protease after cleavage from its pro-domain. Can the authors distinguish BACE1 substrates from ADAM22 substrates?**

ADAM22 belongs to the family of 'A disintegrin and metalloproteinases' (ADAM) proteases. Despite this name, this family does not only contain proteolytically active, but also proteolytically inactive family members, which do not contain the consensus active site motif HEXXHXXGXXH and thus, are not able to act as proteases. ADAM22 is one of the catalytically inactive ADAM family members (Fukata et al., 2006, Gødde et al., 2006, Hsia et al., 2019, Sagane et al., 2005). Thus, your important remark is not relevant in the context of this BACE1 inhibitor experiment.

We added the words "the proteolytically inactive" in front of ADAM22 in the results' section. The new sentence starting in line 261 now is: "The proteolytically inactive ADAM22, which is a new BACE1 substrate candidate, and CD200, which was previously suggested as a BACE1 substrate candidate in a peripheral cell line (Stutzer et al., 2013), were further validated as neuronal BACE1 substrates by Western blots and ELISA assays (Fig. 5C-E).

- 2.3. **In Figure 5c, there are arrows pointing to two bands for flADAM22. The datasheet for the corresponding anti-ADAM22 antibody used does show a predominant single band and then a fainter lower band beneath. Full length CD200 also appears to run as two bands. Can the authors explain why there are two bands for flADAM22/flCD200 and whether or not reducing agent was added to these samples? There are also arrows pointing to the flADAM22 bands in Fig. 5c and there should be a corresponding description in the figure caption or main text.**

The reducing agent β -mercaptoethanol was added to all samples analyzed in this paper by SDS-PAGE.

ADAM22:

In Figure 5C, the two arrows pointing towards flADAM22 indicate the bands corresponding to the mature and immature (or precursor) form of ADAM22. In general, ADAM family members, including the proteolytically active ADAM10 and ADAM17 (Brummer et al., 2018), reveal this typical band pattern in Western blot analysis. ADAMs are produced in an immature form (upper band) that contains the prodomain. While passing through the secretory pathway they

mature by losing their prodomain e.g. due to furin cleavage, resulting in a loss of molecular weight, as seen in immunoblots. This propeptide is annotated in UniProt for the human sequence (accession number Q9P0K1) and was described previously for ADAM22 (Fukata et al., 2006, Gödde et al., 2006). Because the propeptide removal does not happen autocatalytically, but by furin or related proteases, the propeptide removal is seen for both the proteolytically active and the inactive ADAM family members. Importantly, upon BACE inhibitor C3 treatment it is the lower band (mature form of ADAM22) which accumulates, because it is this form that reaches the cellular compartments where BACE1 is active.

CD200

For CD200 there is no evidence of a propeptide, in contrast to ADAM22. However, many membrane proteins, including APP exist in different isoforms, depending on their glycosylation status. Upon protein synthesis and integration into the membrane of the endoplasmic reticulum, the proteins – if N-glycosylated – are modified with “immature”, mannose-rich sugars. This is typically the lower molecular weight band (also seen for APP). As the proteins travel through the Golgi towards the plasma membrane, the mannose-rich sugars are first trimmed and then replaced by so-called complex sugars that add molecular weight to the protein, resulting in a higher molecular weight band. CD200 is intensively N-glycosylated. Thus, the dual band pattern for CD200 is very similar to many other membrane proteins and might therefore be explained by different forms of glycosylation and/or the presence of different isoforms as suggested by Chen et al. 2008.

2.4. **Minor comments: iBAQ is used in Figure 1D and should be defined in the main text.**

iBAQ was defined as following in the manuscript (page 7 and 33):

“iBAQ values representing a rough estimate of molar protein abundances within a sample.”

“The iBAQ values roughly correlate with molar abundance of the proteins. Therefore, a difference of one in log10 scale represents a 10-fold abundance difference.”

2.5. **Can the authors comment on the glycosylation differences between the different cell types? Specifically, do the authors know if there are more or less sialic acid sugars that may allow different amounts of azido-labeling and how this may impact their datasets**

We agree with the reviewer that glycosylation may differ between the diverse cell types of the brain. Systematic studies addressing the variations of glycosylation in the different brain cell types are lacking to date. Thus, it appears possible that the four different cell types have different degrees of sialylation. However, all four cell types are able to sialylate proteins. A recent study by Xie et al studied sialylation *in vivo* and established a list of sialylated proteins. Interestingly, among others were CADM4, TF, MUG1, CTSB and NEO1 on the list of sialylated proteins which were all found to be secreted from all four brain cell types in our hiSPECS secretome resource. Thus, we are confident that hiSPECS is suitable to study the secretome of all brain cell types. We are aware that hiSPECS – similar to other large-scale targeted methods – has limitations, which we mention in the discussion section. We had already written that non-glycosylated, secreted protein will not be detected. Now we also include “non-sialylated secreted protein”. The new sentence (starting on line 417) is: “The secretome of the four major brain cell types and the *ex vivo* tissue identified here represents a snapshot of the total brain secretome and more secreted proteins are known or likely to exist. These include non-glycosylated and non-sialylated secreted proteins which are not captured with hiSPECS as well as proteins secreted in a time-dependent manner such as during development or aging.”

2.6. It is interesting that there is a significantly higher proportion of membrane proteins in the list of unique proteins specific to neuronal cells and that this is not the case for the other three cell types. The authors suggest that this is due to different shedding activity levels. As noted by Sharma et al 2015, these four cell types all have a unique, enriched pattern of integral membrane proteins. The authors also comment that integral membrane proteins are not part of their shedded dataset, consistent with the observation that they did not see the same enrichment. Are there other proteomic studies that have characterized the membrane proteomes for these cell lines? If so, are there differences in the abundance and diversity of membrane proteins that the authors are aware of among these cells.

Unfortunately, there are no membrane or surface proteome data available comparing primary astrocytes, microglia, neurons and oligodendrocytes. Currently, the best data set is indeed the Sharma et al. 2015 paper, as noted by this reviewer.

We would like to clarify one point that we may not have explained well enough and that the reviewer may have also referred to. What we identify in the secretome are secreted proteins that are annotated in UniProt for example as soluble or transmembrane proteins. Are the transmembrane proteins really full-length transmembrane proteins (including their transmembrane and cytoplasmic domains) in the secretome? If yes, these full-length proteins would need to have been released for example from dying cells that break open. To exclude such possibilities we therefore tested whether those secretome proteins, which are annotated as transmembrane proteins, are indeed found in the secretome as full-length proteins or rather as their shed ectodomains. In agreement with our expectation we observed basically only ectodomains and therefore conclude that they result from proteolytic ectodomain shedding.

Reviewer #3

This paper describes an important technical advance that enables analysis of secreted glycoproteins in cultured cells by proteomics. I think this work is exciting and topical, but feel that there are key questions that I as a referee with a non-proteomics background can only partly comment on.

3.1. Is the development of hiSPECS a highly significant advance compared to the previously described SPECS method, such that it qualifies for publication in EMBO J.? I cannot assess this question but a proteomics expert might be able to comment.

Our study provides major improvements of the previous method. This includes increased sensitivity, speed of the method and miniaturization (less material required), so that now for example the secretome of microglia from single mice is amenable. The major technical improvements of the method is only one part of our study. Additionally, we provide a resource of the brain secretome, which can have impact in many different research fields, including in neuroscience. Furthermore, we used the resource to determine the cell type of origin of CSF proteins and proteins secreted under inflammatory conditions from brain slices, which so-far was not systematically possible. Finally, we apply the new hiSPECS method for identification and validation of new BACE1 substrates. Taken together, our manuscript starts out with development of the improved method and its application to set up a resource, which in turn is used to address central questions in neurobiology.

- 3.2. **Is the present paper as a pure technical report appropriate for EMBO J.? This is more an editorial than a scientific question, but I note that the previous study on SPECS by the same authors, published in EMBO J., identified BACE substrates as a biological application whereas the current paper lacks a major biological theme.**

In line with the decision letter of the editor, this manuscript is considered as a resource and demonstrates distinct applications of the resource.

- 3.3. **What interesting new findings does the current study report? The present paper demonstrates that neurons but not other cell types appear to produce a secretome that includes a large number of ECDs from cleaved cell surface proteins. This is interesting, as it suggests that surface proteases are much more active in neurons than other cell types. If the authors could dig a bit deeper on this theme and obtain independent evidence for this observation as well as explore its mechanism and physiological significance, this paper would be much more appropriate for EMBO J. than in its current version.**

Systematic analysis of the iSPECS secretome resource allowed us to pinpoint ectodomain shedding to be a more dominant process in neurons compared to glia cells (see. Figure 3 C,D). We agree that exploring mechanism and determining protease activity levels shaping this imbalance between the different brain cell types is an important question. Given that our manuscript is submitted as a resource, such additional experiments are planned for future studies, in line with the decision letter of the editor.

- 3.4. **My overall impression is that this interesting paper would benefit from a deeper exploration of the biology. I hope my comments will be helpful.**

In line with the decision letter of the editor, this manuscript is considered as a resource and does not require further experiments into the biology.

References

- Bruderer R, Bernhardt OM, Gandhi T, Miladinović SM, Cheng L-Y, Messner S, Ehrenberger T, Zanotelli V, Butscheid Y, Escher C, Vitek O, Rinner O, Reiter L (2015) Extending the limits of quantitative proteome profiling with data-independent acquisition and application to acetaminophen-treated three-dimensional liver microtissues. *Molecular & cellular proteomics : MCP* 14: 1400-1410
- Brummer T, Pignoni M, Rossello A, Wang H, Noy PJ, Tomlinson MG, Blobel CP, Lichtenthaler SF (2018) The metalloprotease ADAM10 (a disintegrin and metalloprotease 10) undergoes rapid, postlysis autocatalytic degradation. *FASEB journal : official publication of the Federation of American Societies for Experimental Biology* 32: 3560-3573
- Fukata Y, Adesnik H, Iwanaga T, Brecht DS, Nicoll RA, Fukata M (2006) Epilepsy-related ligand/receptor complex LGI1 and ADAM22 regulate synaptic transmission. *Science (New York, NY)* 313: 1792-5
- Gillet LC, Navarro P, Tate S, Rost H, Selevsek N, Reiter L, Bonner R, Aebersold R (2012) Targeted data extraction of the MS/MS spectra generated by data-independent acquisition: a new concept for consistent and accurate proteome analysis. *Molecular & cellular proteomics : MCP* 11: O111.016717
- Gödde NJ, D'Abaco GM, Paradiso L, Novak U (2006) Efficient ADAM22 surface expression is mediated by phosphorylation-dependent interaction with 14-3-3 protein family members. *Journal of cell science* 119: 3296-305
- Gosselin D, Skola D, Coufal NG, Holtman IR, Schlachetzki JCM, Sajti E, Jaeger BN, O'Connor C, Fitzpatrick C, Pasillas MP, Pena M, Adair A, Gonda DD, Levy ML, Ransohoff RM, Gage FH, Glass CK (2017) An environment-dependent transcriptional network specifies human microglia identity. *Science (New York, NY)* 356
- Hirano M, Goldman JE (1988) Gliogenesis in rat spinal cord: evidence for origin of astrocytes and oligodendrocytes from radial precursors. *Journal of neuroscience research* 21: 155-67
- Hsia HE, Tushaus J, Brummer T, Zheng Y, Scilabra SD, Lichtenthaler SF (2019) Functions of 'A disintegrin and metalloproteases (ADAMs)' in the mammalian nervous system. *Cellular and molecular life sciences : CMLS* 76: 3055-3081
- Ludwig C, Gillet L, Rosenberger G, Amon S, Collins BC, Aebersold R (2018) Data-independent acquisition-based SWATH-MS for quantitative proteomics: a tutorial. *Molecular systems biology* 14: e8126
- Sagane K, Hayakawa K, Kai J, Hirohashi T, Takahashi E, Miyamoto N, Ino M, Oki T, Yamazaki K, Nagasu T (2005) Ataxia and peripheral nerve hypomyelination in ADAM22-deficient mice. *BMC neuroscience* 6: 33
- Schwanhäusser B, Busse D, Li N, Dittmar G, Schuchhardt J, Wolf J, Chen W, Selbach M (2011) Global quantification of mammalian gene expression control. *Nature* 473: 337-42
- Sebastian Monasor L, Müller SA, Colombo AV, Tanrioever G, König J, Roth S, Liesz A, Berghofer A, Piechotta A, Prestel M, Saito T, Saido TC, Herms J, Willem M, Haass C, Lichtenthaler SF, Tahirovic S (2020) Fibrillar A β triggers microglial proteome alterations and dysfunction in Alzheimer mouse models. *eLife* 9
- Sharma K, Schmitt S, Bergner CG, Tyanova S, Kannaiyan N, Manrique-Hoyos N, Kongi K, Cantuti L, Hanisch UK, Philips MA, Rossner MJ, Mann M, Simons M (2015) Cell type- and brain region-resolved mouse brain proteome. *Nature neuroscience* 18: 1819-31
- Stiess M, Wegehngel S, Nguyen C, Nickel W, Bradke F, Cambridge SB (2015) A Dual SILAC Proteomic Labeling Strategy for Quantifying Constitutive and Cell-Cell Induced Protein Secretion. *Journal of proteome research* 14: 3229-3238
- Stutzer I, Selevsek N, Esterhazy D, Schmidt A, Aebersold R, Stoffel M (2013) Systematic proteomic analysis identifies beta-site amyloid precursor protein cleaving enzyme 2 and 1 (BACE2 and BACE1) substrates in pancreatic beta-cells. *The Journal of biological chemistry* 288: 10536-47
- Voytyuk I, Mueller SA, Herber J, Snellinx A, Moechars D, van Loo G, Lichtenthaler SF, De Strooper B (2018) BACE2 distribution in major brain cell types and identification of novel substrates. *Life Sci Alliance* 1: e201800026-e201800026

Dear Stefan,

Thank you for submitting your revised manuscript to The EMBO Journal. I am sorry for the delay in getting back to you, but I have been away for a few days and also wanted to take the time to carefully look through everything.

I have now looked at your response and I appreciate the introduced changes. I am therefore very pleased to accept the manuscript for publication here. Before sending you the formal accept letter there are just a few editorial changes that need to be done.

- In the author contribution list there are 2 MS listed - please distinguish between MSu and MSi.
- Please upload the synopsis image and text as a separate files. Would be good to have the synopsis image as a TIFF/JPEG file.
- Please relabel Structured Methods as Materials and Methods. Please also double check that the MS sections are in the right order
- The EV table legends need to be removed from the Article file.
- We encourage the publication of source data, particularly for electrophoretic gels and blots, with the aim of making primary data more accessible and transparent to the reader. It would be great if you could provide me with a PDF file per figure that contains the original, uncropped and unprocessed scans of all or key gels used in the figure? The PDF files should be labeled with the appropriate figure/panel number, and should have molecular weight markers; further annotation could be useful but is not essential. The PDF files will be published online with the article as supplementary "Source Data" files.
- I have asked our publisher to do their pre-publication checks on the paper. They will send me the file within the next few days. Please wait to upload the revised version until you have received their comments.

That should be all and congratulations on a nice study!

best Karin

Karin Dumstrei, PhD
Senior Editor
The EMBO Journal

When assembling figures, please refer to our figure preparation guideline in order to ensure proper

formatting and readability in print as well as on screen:

Further information is available in our Guide For Authors:

The revision must be submitted online within 90 days; please click on the link below to submit the revision online before 5th Nov 2020.

Link Not Available

The authors performed the requested editorial changes.

Dear Stefan,

Thank you for submitting your revised manuscript to The EMBO Journal. I have now had a chance to take a careful look at everything and all looks good.

I am therefore very pleased to accept the manuscript for publication here.

Congratulations on a nice study!

best Karin

Karin Dumstrei, PhD
Senior Editor
The EMBO Journal

Please note that it is EMBO Journal policy for the transcript of the editorial process (containing referee reports and your response letter) to be published as an online supplement to each paper. If you do NOT want this, you will need to inform the Editorial Office via email immediately. More information is available here: http://emboj.embopress.org/about#Transparent_Process

Your manuscript will be processed for publication in the journal by EMBO Press. Manuscripts in the PDF and electronic editions of The EMBO Journal will be copy edited, and you will be provided with page proofs prior to publication. Please note that supplementary information is not included in the proofs.

Should you be planning a Press Release on your article, please get in contact with embojournal@wiley.com as early as possible, in order to coordinate publication and release dates.

If you have any questions, please do not hesitate to call or email the Editorial Office. Thank you for your contribution to The EMBO Journal.

Corresponding Author Name: Stefan F. Lichtenthaler

Manuscript Number: EMBOJ-2020-105693